# Error Discovery By Clustering Influence Embeddings

**Fulton Wang**[*]
Meta

**Julius Adebayo**[*]
Prescient Design / Genentech

**Sarah Tan**
Cornell University

**Diego Garcia-Olano**
Meta

**Narine Kokhlikyan**
Meta

## Abstract

We present a method for identifying groups of test examples—slices—on which a model under-performs, a task now known as *slice discovery*. We formalize *coherence*—a requirement that erroneous predictions, within a slice, should be *wrong for the same reason*—as a key property that any slice discovery method should satisfy. We then use influence functions to derive a new slice discovery method, `InfEmbed`, which satisfies coherence by returning slices whose examples are influenced similarly by the training data. `InfEmbed` is simple, and consists of applying K-Means clustering to a novel representation we deem *influence embeddings*. We show `InfEmbed` outperforms current state-of-the-art methods on 2 benchmarks, and is effective for model debugging across several case studies.[2]

## 1 Introduction

Error analysis is a longstanding challenge in machine learning [Pope, 1976, Amershi et al., 2015, Sculley et al., 2015, Chung et al., 2019, Chakarov et al., 2016, Cadamuro et al., 2016, Kearns et al., 2018, Kim et al., 2019, Zinkevich Martin, 2020]. Recently, Eyuboglu et al. [2022b] formalized a type of error analysis termed the *slice discovery problem*. In the *slice discovery problem*, given a multi-class classification model and test data, the goal is to partition the *test* data into a set of *slices*—groups of test examples—by model performance. An effective slice discovery method (SDM) should satisfy two desiderata:

1. *Error surfacing*: Identify *under-performing slices*, i.e. slices with low accuracy, if they exist.
2. *Coherence*: Return slices that are *coherent* in the following sense: Erroneous predictions in a given slice should have the same *root cause*, i.e. be "wrong for the same reason".

To illustrate, suppose we have a pre-trained model that detects hip fractures from an x-ray, trained on data from one hospital, that is now deployed in a new hospital. The model might err frequently on two groups: 1) examples containing a spurious signal like the frequency signature of the scanner from the first hospital [Badgeley et al., 2019], and 2) examples from patients not represented in the new hospital. An effective SDM should uncover these two groups. The slice discovery problem has taken on renewed importance given that models trained via empirical risk minimization exhibit degraded performance on certain data groups—slices—due to distribution shift [Koh et al., 2021], noisy labels [Northcutt et al., 2021], and imbalanced data [Idrissi et al., 2022].

In developing a SDM, there are two key challenges. First, *formalizing the coherence property*, i.e., defining the *root cause* of an error, identifying whether two errors have similar root causes, and ultimately, defining whether a slice is *coherent*. Second, *identifying the representation to use*:

---

[*]The two authors contributed equally. Correspondence to fultonwang@meta.com

[2]Code to replicate our findings is available at: https://github.com/adebayoj/infembed

37th Conference on Neural Information Processing Systems (NeurIPS 2023).

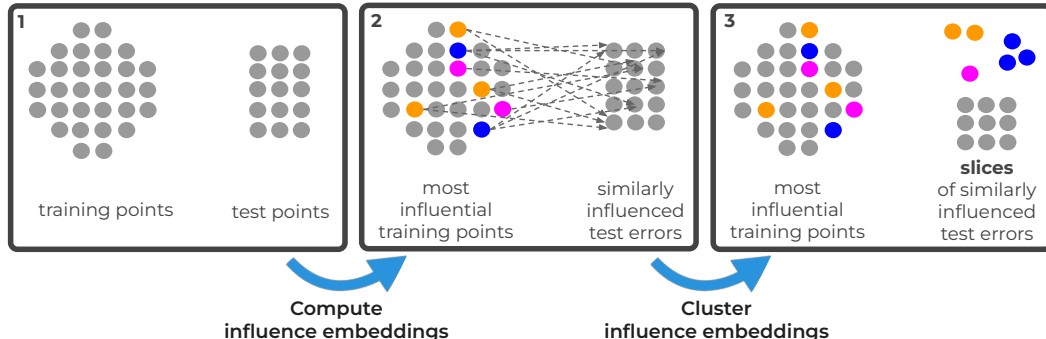

Figure 1: Schematic of the `InfEmbed` slice discovery method. In slice discovery, the goal is to partition the test points into groups—slices—by model performance. First, for each test point, we compute its influence embeddings—a low dimensional representation of the vector of influence scores of the training set. In the second stage, we cluster this influence embedding set to group points by training set influence.

returning to the x-ray example, there is typically no additional metadata that indicates whether an x-ray contains the spuriously correlated scanner pattern. The input to a SDM will typically be the x-ray or a representation thereof, without additional metadata that indicates which features are spurious. Thus, identifying a representation that can help distinguish under-performing groups from high performing ones while maintaining coherence is a key challenge.

**Influence Explanations**: To formalize the coherence property, we appeal to influence functions [Koh and Liang, 2017], which tractably estimate the "effect" of a training example on a test example—how much the loss of the test example would change if the model were re-trained without that training example. We define the *influence explanation* of a given test example to be the vector of the 'influence' of each training example on the test example. Since a test example's influence explanation is the "cause" of its prediction from a training dataset perspective, we say a SDM achieves *coherence* if the slices it returns have similar influence explanations.

**Influence Embeddings**: To create coherent slices, one might naively cluster test examples' *influence explanations*. However, each influence explanation is high dimensional—equal to the size of the training dataset. Instead, we transform influence explanations into *influence embeddings*. We define influence embeddings to be vectors with the *dot-product property*: that the influence of a training example on a test example is the dot-product of their respective embeddings.

**Clustering Influence Embeddings**: Given the coherence-inducing properties of influence embeddings, we propose `InfEmbed`, an SDM that applies K-means clustering to the influence embeddings of all test examples. Despite its simplicity, `InfEmbed` satisfies the two aforementioned desiderata. First, it is error-surfacing, as mis-predicted examples that are "wrong for the same reasons", i.e. have similar influence explanations, tend to be grouped into the same slice. Second, `InfEmbed` satisfies several desirable properties. It produces coherent slices, and can accomodate user-defined choices regarding the subset of a model's representations to use, ranging from all to only the last layer [d'Eon et al., 2022, Sohoni et al., 2020]. In addition, the slices produced by `InfEmbed` tend to have *label homogeniety*, i.e. examples are homogeneous in terms of the true and predicted label. Unlike other SDMs, these properties are implicit consequences the rigorous influence function-based derivation.

**Contributions**: We summarize our contributions as follows:

1. **New slice discovery method**: We propose `InfEmbed`, a SDM to identify under-performing groups of data that applies K-Means to a new representation: *influence embeddings*.

2. **Coherence & Theoretical Justification.** We formalize the *coherence* property, which requires that erroneous predictions within a slice should be "wrong for the same reason". We then define influence embeddings, and prove that a procedure which clusters them achieves coherence: it returns slices whose examples are influenced similarly by the training data.

3. **Empirical Performance.** We show that `InfEmbed` outperforms previous state-of-the-art SDMs on 2 benchmarks—DCBench [Eyuboglu et al., 2022a], and SpotCheck [Plumb et al., 2022]. `InfEmbed` is scalable, able to recover known errors in variety of settings, and identifies coherent slices on realistic models.

## 2 Background

At a high level, the goal of the slice discovery problem is to partition a test dataset into slices–groups of data points—such that some slices are *under-performing*, i.e. have low accuracy, and slices are *coherent*, i.e. erroneous predictions in a given slice are "wrong for the same reason". We first formally define the slice discovery problem, and then overview influence functions, which we use to define what it means for a slice to be coherent - one of our key contributions, and which will be detailed in Section 3.

**Slice Discovery Problem:** Given a trained *multi-class* classification model $f(\cdot; \theta)$ over $C$ classes with model parameters $\theta$ and an example $x$ from some input space $X$, $f(x; \theta) \in \mathbb{R}^C$ is the prediction for the example and $f(x; \theta)_c$ is the pre-softmax prediction for class $c$. Given a test dataset $\mathbf{Z} \coloneqq [z_1, ..., z_N]$ with $N$ examples where $z_i \coloneqq (x_i, y_i)$ and $x_i \in \mathcal{X}, y_i \in \mathcal{Y} \coloneqq [0,1]^C$, the goal of the slice discovery problem is to partition the test dataset into $K$ slices: $\{\Phi_k\}_{k=1}^K$, where $\Phi_k \subseteq [N]$, $\Phi_k \cap \Phi_{k'} = \emptyset$ for $k \neq k'$, $\cup_{k=1}^K \Phi_k = [N]$. As we will make precise shortly, each slice $\Phi_k$ should be coherent, and some slices should be under-performing slices. The model is assumed to be trained on a training dataset with $N'$ examples, $\mathbf{Z}' \coloneqq [z'_1, ..., z'_{N'}]$ where $z'_i \coloneqq (x'_i, y'_i)$ with $x'_i \in \mathcal{X}, y'_i \in \mathcal{Y}$, so that $\theta = \operatorname{argmin}_{\theta'} \frac{1}{N'} \sum_{i=1}^{N'} L(z'_i; \theta')$. We assume the loss function $L$ is cross-entropy loss. Thus, given test point $z = (x, y)$ with $x \in \mathcal{X}, y = [y_1, ..., y_C] \in \mathcal{Y}$, $L(z; \theta) = -\sum_c y_c \log p_c$, with $p = [p_1, ..., p_C]$ and $p_c \coloneqq \frac{\exp(f(x;\theta)_c)}{\sum_{c'} \exp(f(x;\theta)_{c'})}$, i.e. $p = \operatorname{softmax}(f(x; \theta))$.

**Influence Functions**: Influence functions estimate the *effect* of a given training example, $z'$, on a test example, $z$, for a pre-trained model. Specifically, the influence function approximates the change in loss for a given test example $z$ when a given training example $z'$ is removed from the training data and the model is retrained. Koh and Liang [2017] derive the aforementioned influence to be $I(z', z) \coloneqq \nabla_\theta L(z'; \theta)^\intercal H_\theta^{-1} \nabla_\theta L(z; \theta)$, where $H_\theta$ is the loss Hessian for the pre-trained model: $H_\theta \coloneqq 1/n \sum_{i=1}^n \nabla_\theta^2 L(z; \theta)$, evaluated at the pre-trained model's final parameter checkpoint. The loss Hessian is typically estimated with a random mini-batch of data.

The main challenge in computing influence is that it is impractical to explicitly form $H_\theta$ unless the model is small, or if one only considers parameters in a few layers. Schioppa et al. [2022] address this problem by forming a low-rank approximation of $H_\theta^{-1}$ via a procedure that does not explicitly form $H_\theta$. In brief, they run the Arnoldi iteration [Trefethen and Bau III, 1997] for $P$ iterations to get a $P$-dimensional Krylov subspace for $H_\theta$, which requires only $P$ Hessian-vector products. Then, they find a rank-$D$ approximation of the restriction of $H_\theta$ to the Krylov subspace, a small $P \times P$ matrix, via eigendecomposition. Their algorithm, FactorHessian (see Appendix B for details), which we use in our formulation, returns factors of a low-rank approximation of $H_\theta^{-1}$

$$M, \lambda = \operatorname{FactorHessian}(\mathbf{Z}', \Theta, P, D) \text{ where} \tag{1}$$

$$\hat{H}_\theta^{-1} \coloneqq M\lambda^{-1}M^\intercal \approx H_\theta^{-1}, \tag{2}$$

$M \in \mathbb{R}^{|\theta| \times D}$, $\lambda \in \mathbb{R}^{D \times D}$ is a *diagonal* matrix, $|\theta|$ is the parameter count, $\hat{H}_\theta^{-1}$ approximates $H_\theta^{-1}$, $D$ is the rank of the approximation, $P$ is the Arnoldi dimension, i.e. number of Arnoldi iterations, and here and everywhere, *configuration* $\Theta \coloneqq (L, f, \theta)$ denotes the loss function, model and parameters. $\hat{H}_\theta^{-1}$ is then used to define the *practical influence* of training example $z'$ on test example $z$:

$$\hat{I}(z', z) \coloneqq \nabla_\theta L(z'; \theta)^\intercal \hat{H}_\theta^{-1} \nabla_\theta L(z; \theta). \tag{3}$$

## 3 Error Discovery By Clustering Influence Embeddings

We propose a slice discovery method (SDM) whose key contribution is to require returned slices have *coherence*, and unlike past works, rigorously define coherence, which we do using influence functions. In this section, we derive our SDM as follows: 1) We formally define that a slice has coherence if its examples have similar *influence explanations*—which we define using influence functions, and leverage them to formalize a clustering problem. 2) Because influence explanations are high-dimensional, we derive *influence embeddings*, a low-dimensional similarity-preserving approximation. 3) Using influence embeddings, we give a simple and efficient procedure to solve the clustering problem to give our SDM, `InfEmbed`. 4) We then describe an extension, `InfEmbed-Rule`, which instead of returning a partition with a user-specified number of slices, returns the largest slices

satisfying user-specified rules. 5) We then show that the proposed formulation implicitly encourages a property, label homogeneity, that previous methods needed to explicitly encourage, in a manner that requires specifying a hard-to-tune hyperparameter. 6) Lastly, explain slices using training samples.

## 3.1 Influence Explanations and Problem Formulation

We seek to partition the test data into slices such that predictions for examples in the same slice have similar root causes as quantified by influence functions. Therefore, we propose to represent the root cause of the prediction for a test example, $z$, with its *influence explanation $E(z)$*:

$$E(z) := \{\hat{I}(z'_j, z)\}_{j=1}^{N'}, \tag{4}$$

the vector containing the influence of every training example on the test example. Thus, test examples with similar influence explanations are "influenced similarly by the training data". We then want each slice be *coherent* in the following sense: the Euclidean distance between influence explanations of examples in the same slice should be low.

---

**Algorithm 2** Our SDM, `InfEmbed`, applies K-Means to influence embeddings of test examples.

1: **procedure** GETEMBEDDINGS($\mathbf{Z}, \mathbf{Z}', \Theta, P, D$)
2:     **Outputs**: $D$-dimensional embeddings for test data $\mathbf{Z}$
3:     $M, \lambda \leftarrow$ FactorHessian($\mathbf{Z}', \Theta, P, D$)             ▷ see Equation 2
4:     $\mu_i \leftarrow \lambda^{-1/2} M^\intercal \nabla_\theta L(z_i; \theta)$ for $i = 1, \ldots, N$     ▷ see Equation 6
5:     $\boldsymbol{\mu} \leftarrow [\mu_1, \ldots, \mu_N]$
6:     **Return:** $\boldsymbol{\mu}$
7: **end procedure**
8: **procedure** INFEMBED(K, $\mathbf{Z}, \mathbf{Z}', \Theta, P, D$)
9:     **Inputs**: number of slices $K$, training dataset $\mathbf{Z}'$, test dataset $\mathbf{Z}$, configuration $\Theta$, Arnoldi dimension $P$, influence embedding dimension $D$
10:     **Outputs**: partition of test dataset into slices, $\Phi$
11:     $\boldsymbol{\mu} \leftarrow$ GetEmbeddings($\mathbf{Z}, \mathbf{Z}', \Theta, P, D$)
12:     $[r_1, \ldots, r_N] \leftarrow$ K-Means($\boldsymbol{\mu}, K$)     ▷ compute cluster assignments for all $N$ examples
13:     **Return:** $\{\{i \in [N] : r_i = k\}$ for $k \in [K]\}$     ▷ convert cluster assignments to a partition
14: **end procedure**

---

Therefore, we solve the slice discovery problem described in Section 2 via the following formulation—return a partition of the test dataset into slices, $\{\Phi_k\}_{k=1}^K$, that minimizes the total Euclidean distance between influence explanations of examples in the same slice. Formally, we seek:

$$\text{argmin}_{\{\Phi_k\}_{k=1}^K} \sum_k \sum_{i,i' \in \Phi_k} ||E(z_i) - E(z_{i'})||^2. \tag{5}$$

## 3.2 Influence Embeddings to approximate Influence Explanations

To form coherent slices, a naive approach would be to form the influence explanation of each test example, and apply K-Means clustering with Euclidean distance to them. However, influence explanations are high-dimensional, with dimensionality equal to the size of the training data, $N'$. Instead, we will use *influence embeddings*: vectors such that the practical influence of a training example on a test example is the dot-product of their respective influence embeddings. Looking at Equations 2 and 3, we see the influence embedding $\mu(z)$ of an example $z$ must be defined as

$$\mu(z) := \lambda^{-1/2} M^\intercal \nabla_\theta L(z; \theta) \tag{6}$$

and $M, \lambda$ are as defined in Section 2, because for any training example $z'$ and test example $z$, $\hat{I}(z', z) = \mu(z')^\intercal \mu(z)$. Procedure `GetEmbeddings` of Algorithm 1 finds $D$-dimensional influence embeddings for the test dataset $\mathbf{Z}$, which has runtime *linear* in the Arnoldi dimension $P$. The rank $D$ of the factors from `FactorHessian` determines the dimension of the influence embeddings.

Influence embeddings satisfy a critical property—that if two examples have similar influence embeddings, they also tend to have similar influence explanations. This is formalized by the following lemma, whose proof follows from the Cauchy-Schwartz inequality, and that influence is the dot-product of influence embeddings (see Appendix Section B.1)

**Lemma 1:** There is a constant $C > 0$ such that for any test examples $z_i, z_j$, $||E(z_i) - E(z_j)||^2 \leq C||\mu(z_i) - \mu(z_j))||^2$.

### 3.3 `InfEmbed`: Discovering Problematic Slices by Clustering Influence Embeddings

We will solve the formulation of Equation 5 via a simple procedure which applies K-Means with Euclidean distance to the influence *embeddings* of the test dataset, $\boldsymbol{\mu} \coloneqq [\mu(z_1), \ldots, \mu(z_N)]$. This procedure is justified as follows: Using Lemma 1, given any partition $\{\Phi_k\}$, we know

$$\sum_k \sum_{i,i' \in \Phi_k} ||E(z_i) - E(z_{i'})||^2 \leq C \sum_k \sum_{i,i' \in \Phi_k} ||\mu(z_i) - \mu(z_{i'})||^2. \qquad (7)$$

However, the quantity in the right of Equation 7 is the *surrogate* objective minimized by this procedure—it is the K-Means objective applied to influence embeddings, scaled by $C > 0$ which does not matter. The quantity in the left of Equation 7 is the *actual* objective which the formulation of Equation 5 minimizes. Therefore, the procedure is minimizing a surrogate objective which upper bounds the actual objective we care about. Put another way, by applying K-Means to influence embeddings, examples within the same slice will not only have similar influence embeddings, but also similar influence explanations as desired, by Lemma 1. Algorithm 1 describes `InfEmbed`. In addition, we find that normalizing the centroid centers in each iteration of K-Means is helpful, perhaps due to lessening the effect of outlier influence embeddings. Overall, instead of K-means, other clustering algorithms can used as part of formulation.

**Computational Complexity**: We now take a step back to examine the computational complexity of the proposed procedure. The main computational bottleneck is the implicit Hessian estimation required to compute influence embeddings. Schioppa et al. [2022]'s approach, which we rely on, has complexity $\mathcal{O}(P)$ where $P$ is the Arnoldi dimension. This is because each of the $P$ steps in the Arnoldi iteration requires computing a Hessian-vector product. Consequently, the complexity of computing influence embeddings is exactly the same as that of the influence function (IF) estimation. For the K-means portion, the complexity is $\mathcal{O}$(number of samples$\times$ number of k-means iterations $\times$ number of clusters). In practice, the clustering step is near instantaneous, so the method is dominated by implicit Hessian estimation step.

### 3.4 `InfEmbed-Rule`: Finding Slices Satisfying a Rule

The key hyper-parameter of the $\mathrm{InfEmbed}$ method is $K$, the number of slices to return. In practice, it may not be intuitive for a user to choose $K$. Instead, the user may want to know if there exist any coherent slices that are problematic, where 'problematic' is pre-specified by a rule, e.g., *a slice with greater than T samples and that has accuracy less than A*. Without specifying the rule, a practitioner has to iterate through all the returned slices to figure out which one is problematic. Therefore we propose a procedure that recursively clusters influence embeddings until slices satisfying a pre-specified rule are found, or until the slices are too small.

The proposed approach, `InfEmbed-Rule`, is analogous to building a tree to identify slices satisfying the rule, where the splits are determined by K-Means clustering of influence embeddings. In addition to letting the user specify more intuitive hyperparameters, this procedure also has the advantage that if a large slice with sufficiently low accuracy is found, it will not be clustered further. Appendix Algorithm B.2 outlines this `InfEmbed-Rule` method. Its inputs are the same as `InfEmbed`, except instead of $K$, one specifies accuracy threshold $A$, size threshold $S$, and branching factor $B$, which sets how many clusters the K-means call in each step of the recursion should return. It outputs a set of slices each with accuracy less than $A$ and size greater than $S$.

### 3.5 Properties of Discovered Slices

We now show that the slices discovered by `InfEmbed` possess properties similar to past SDM's. To derive these properties, we consider what factors result in two examples $z = (x, y)$, $z' = (x', y')$ having influence embeddings with low Euclidean distance, so that K-Means would place them in the same slice. To simplify analysis, we note that two examples would be placed in the same slice if their *gradients* have high dot-product, because influence embeddings are linearly transformed gradients (see Equation 6) and Euclidean distance is equal to their negative dot-product plus a constant. For further simplicity, we consider the case when the model $f$ is a linear model, i.e. when only gradients in the last fully-connected layer are considered when computing influence embeddings. Then, a straightforward computation shows that two examples $z, z'$ will be placed in the same slice if

$$\nabla_\theta L(z; \theta)^\intercal \nabla_\theta L(z'; \theta) = (y - p)^\intercal (y' - p') x^\intercal x', \qquad (8)$$

is high, where $L$ is cross-entropy loss, $\theta$ are the parameters, and for example $z$, $p := [p_1, \ldots, p_C] = \text{softmax}(f(x))$ is the predicted probabilities for each class, $y$ is the one-hot encoded label, $x$ is its last-layer representation, and $p', y', x'$ are defined analogously for example $z'$.

Looking at the $x^\intercal x'$ term of Equation 8, we see that two examples will be placed in the same slice if, all else being equal, their last-layer representations are similar as measured by the dot-product. Past SDM's [d'Eon et al., 2022, Sohoni et al., 2020] also use *heuristic* similarity measures involving last-layer representations to decide if two examples should be in the same slice. On the other hand, our influence function-based derivation shows that their dot-product is the "canonical" similarity measure to use. Note that since influence embeddings consider gradients in layers beyond the last, they are able to consider information *beyond* the last-layer representations, *generalizing* past SDM's.

Looking at the $(y - p)^\intercal (y' - p')$ term of Equation 8, we see that two examples will be placed in the same slice if, all else being equal, their *margins* are similar, as measured by dot-product. The margin of an example, eg. $(y - p)$ or $(y' - p')$, is the vector containing the difference between the true label and predicted probability for each class. Thus, examples with similar labels and predictions will tend to be in the same slice, i.e. slices tend to have *label homogeniety*. Past SDM's [Eyuboglu et al., 2022b] also produce slices that tend to have label homogeniety, but do so through an "error-aware" mixture model that requires choosing a hard-to-tune tradeoff parameter. On the other hand, our influence function-based derivation shows that the dot-product of margins is the "canonical" way to encourage label homogeniety, and does not require tuning a trade-off constant. Note that label homogeniety is *not* the only factor in consideration when forming slices.

### 3.6 Slice Explanation via Slice Opponents

Given an under-performing slice, we are interested in identifying the root-cause of the erroneous predictions in it. We therefore compute the top-$k$ slice opponents and consider them as the root-cause. These are the $k$ training examples whose influence on the slice are most harmful, i.e. whose influence on the total loss over all examples in the slice is the most negative.

Formally, given a slice $\Phi$, we first compute the influence embeddings of the training data, $[\mu(z'_1), \ldots, \mu(z'_{N'})]$. Then, we compute the sum of the influence embeddings of the slice, $v := \sum_{i \in \Phi} \mu(z_i)$. Finally, due to the properties of influence embeddings, the top-$k$ slice opponents of the given slice, $O$, are the $k$ training examples $z'$ for which $v^\intercal \mu(z')$ is the most negative, i.e. $O = \text{argmin}_{O' \subseteq [N'], |O'|=k} v^\intercal (\sum_{i \in O'} \mu(z'_{i'}))$. Inspecting slice opponents can provide insight into the key features responsible for low performance in a slice.

## 4 Results

Here, we perform a quantitative evaluation of `InfEmbed` on the `dcbench` benchmark, and also perform several case studies showing `InfEmbed`'s usefulness. Please see Appendix Section H on a case study applying `InfEmbed` to a small dataset where additional metadata can help explain slices. For scalability, to compute influence embeddings, at times we will only consider gradients in some layers of the model when calculating the gradient $\nabla_\theta L(z; \theta)$ and inverse-Hessian factors $M, \lambda^{1/2}$ in Equation 6. Furthermore, following Schioppa et al. [2022], we do not compute those factors using the entire training data, i.e. we pass in a subset of the training data to `FactorHessian`. For all experiments, we use Arnoldi dimention $P = 500$, and influence embedding dimension $D = 100$, unless noted otherwise. In the experiments that use `InfEmbed-Rule`, we used branching factor B=3. The rationale is that B should not be too large, to avoid unnecessarily dividing large slices with sufficient low accuracy into smaller slices. In practice, B=2 and B=3 did not give qualitatively different results.

### 4.1 InfEmbed on dcbench

**Overview & Experimental Procedure**: `dcbench` [Eyuboglu et al., 2022a] provides 1235 pre-trained models that are derived from real-world data, where the training data is manipulated in 3 ways to lead to test-time errors. This gives 3 tasks - discovering slices whose errors are due to the following manipulations: 1) "rare" (examples in a given class are down-sampled), 2) "correlation" (a spurious correlation is introduced), and 3) "noisy label" (examples in a given class have noisy labels). Since the error causing slice is known a priori, `dcbench` can serve as a way to assess a new SDM. The

underlying datasets for those tasks include 3 input types: natural images, medical images, and medical time-series. Eyuboglu et al. [2022a] also introduced Domino, the SDM that is currently the best-performing SDM on `dcbench`, which uses multi-modal CLIP embeddings as input to an error-aware mixture model. Following Eyuboglu et al. [2022a], we compare `InfEmbed` to Domino using the precision-at-k measure, which measures the proportion of the top k ($k = 10$) elements in the discovered slice that are in the ground truth slice. We evaluate in the setting where errors are due to a trained model.

**Results**: Table 1 compares `InfEmbed` to Domino across the 3 tasks and 3 input types. `InfEmbed` always beats Domino except on the "noisy" task for the EEG Medical Time series input type. We can also compute **coherence scores**—the k-means objective within each cluster—for each setting in the domino benchmark,

We also compute the coherence scores, the K-means objective that we cluster, across all slices for influence embedding clustering as well as for domino. We find a similar correspondence to the previous results. In every setting where we outperform Domino (1219 out of 1235 trained

| SDM Approach | Rare | Correlation | Noisy Label |
|---|---|---|---|
| **Natural Images** | | | |
| Domino | 0.4 | 0.45 | 0.6 |
| Inf-Embed | 0.65 | 0.55 | 0.67 |
| **Medical Images** | | | |
| Domino | 0.39 | 0.6 | 0.58 |
| Inf-Embed | 0.57 | 0.62 | 0.73 |
| **Medical Time Series** | | | |
| Domino | 0.6 | 0.55 | 0.9 |
| Inf-Embed | 0.64 | 0.65 | 0.81 |

Table 1: `InfEmbed` generally outperforms Domino across 3 input types and 3 tasks on the `dcbench` benchmark in terms of precision (higher is better).

settings), our coherence scores are better, in some cases, by almost 50 percent. Regarding label homogeneity: In high performing clusters, we find that influence embedding clustering has lower label homogeneity than domino. However, for error clusters we find the reverse. The goal of our scheme is to find clusters where the model is 'wrong for the same reason'. We conjecture that influence embeddings are most effective for these settings.

## 4.2 `InfEmbed` on the SpotCheck Benchmark

**Overview & Experimental Procedure.** The SpotCheck benchmark Plumb et al. [2022] is based on a synthetic task consisting of 3 semantic features that can be easily controlled to determine the number of 'blindspots' in a dataset. A blindspot is a feature responsible for a model's mistake. Similar to `dcbench`, a model trained on data generated from SpotCheck is induced to make mistakes on an input that has a set of blindspots. The task is to predict the presence of a square in the image. Attributes of

| SDM Approach | DR | FDR |
|---|---|---|
| Barlow | 0.43(0.04) | 0.03(0.01) |
| Spotlight | 0.79(0.03) | 0.09 (0.55) |
| Domino | 0.64 (0.04) | 0.07(0.01) |
| PlaneSpot | 0.85 (0.03) | 0.07 (0.01) |
| Inf-Embed | 0.91 (0.09) | 0.15 (0.62) |

Table 2: Spotcheck Benchmark results.

the input image such as the background, object color, and co-occurrence can be varied to induce mistakes in models obtained from the manipulated data generation process. Plumb et al. [2022] find that current SDM approaches struggle in the presence of several blindspots. Consequently, they propose PlaneSpot, an SDM that uses the representation of the model's last layer, projected to 2 dimensions, along with the model's 'confidence' as part of a mixture model to partition the dataset. They show that PlaneSpot outperforms current approaches as measured by discovery and false discovery rates. We replicate the SpotCheck benchmark, for a subset of features, and compare `InfEmbed` to PlaneSpot. We refer to the Appendix Section D.1 for additional details.

**Results.** We report the performance of the influence embedding approach compared to PlaneSpot on the Spotcheck benchmark in Table 2. We find that Inf-Embed approaches improves upon PlaneSpot's performance on the benchmark, which indicates that the proposed approach is an effective SDM.

**Alternative Clustering Approaches.** To check that the proposed scheme is robust to the choice of clustering algorithm, we replicate the spotcheck experiments with 2 additional clustering algorithms: 1) DBSCAN, and 2) Spectral Clustering. Similar to Table 2, we find that the discovery rate for these algorithms are: 0.92, and 0.89 respectively. In both both cases, the performance of the `InfEmbed`

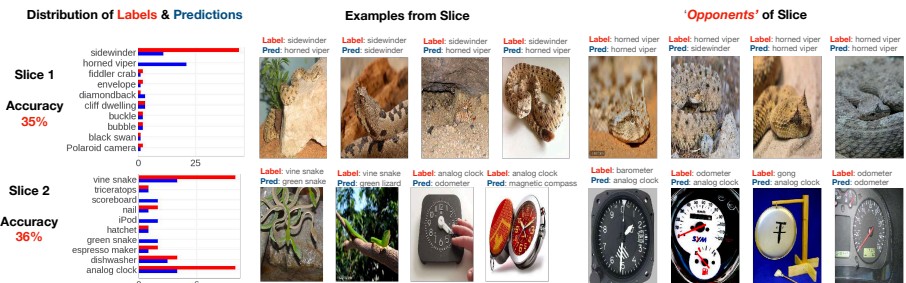

Figure 2: **Overview of Influence embedding Clustering for a Resnet-18 on ImageNet.** We show two slices from applying the influence embedding approach to the ImageNet data. We apply the RuleFind algorithm and identify slices with accuracy under 40 percent and atleast 25 samples. **Left**: On the left, we show the distribution of labels and model predictions in that slice (Labels are in red, and Prediction are blue). **Middle**: Representative examples from the slice. **Right**: We show four opponent examples for the identified slice.

procedure does not degrade, which indicates that the influence embedding representations are robust to the clustering algorithm used.

### 4.3 Imagenet & High Dimensional Settings

**Overview & Experimental Procedure:** Here, we identify problematic slices in a "natural" dataset - the test split of Imagenet [Deng et al., 2009], containing 5000 examples. This image-classification problem is a 1000-class problem, and the analyzed model is a pre-trained Resnet-18 model achieving 69% test accuracy. To compute influence embeddings, we consider gradients in the "fc" and "layer4" layers, which contain 8.9M parameters. We use `ImfEmbed-Rule` to find slices with at most 40% accuracy, and at least 25 examples. To diagnose the root-cause of errors for a slice, we examine the slice's opponents, following Section 3.6.

**Results:** We find 25 slices, comprising a total of 954 examples, whose overall accuracy is 34.9%. Figure 2 displays 2 of these slices (See Section E for additional examples). For each slice, we show the distribution of predicted and true labels, the 4 examples nearest the slice center in influence embedding space, and the 4 strongest opponents of the slice. First, we see that the slices have the label homogeneity property as explained in Section 3.5—the predicted and true labels typically come from a small number of classes. In the first slice, the true label is mostly "sidewinder", and often predicted to be "horned viper". The slice's strongest opponents are horned vipers, hard examples of another class, which the model considers to be similar to sidewinders. In the second slice, we observe *two* dominant labels. Here, vine snakes and analog clocks are often mis-predicted. A priori, one would not have guess the predictions, for these seemingly, unrelated classes would be wrong for the same reasons. However, looking at the opponents, we see all of them have a clock hand, which turns out to be similar to a snake. Also, because the opponents are of a variety of classes, the mis-predictions are also of a variety of classes. Overall, we demonstrate here that our procedure can identify problematic slices with low performance and then trace the cause of the errors to a subset of the training data.

### 4.4 AGNews & Mis-labelled Data

**Overview & Experimental Procedure**: We identify problematic slices in another "natural" dataset - the test split of AGNews [Zhang et al., 2015], comprising 7600 examples. This text-classification problem is a 4-class problem, and the pre-trained model is a BERT base model fine-tuned on the training dataset [3], achieving 93% test accuracy, making 475 errors. We use `ImfEmbed` with $K = 25$, and examine slices with at most 10% accuracy, and at least 10 examples.

**Results**: We find 9 such slices, comprising a total of 452 examples, whose overall accuracy is only 3%. Table 3 shows 2 slices, both of which have label homogeniety. In the first slice, the model achieves 0% accuracy, systematically mis-predicting "business" examples to be "world", perhaps predicting any text with a country name in it to be "world". In the second slice, most labels are

---

[3] https://huggingface.co/fabriceyhc/bert-base-uncased-ag_news

| Slice with 45 examples, 0% accuracy. Predictions: 100% "world", labels: 100% "business" |
|---|
| 1. *"British grocer Tesco sees group sales rise 12.0-percent. Britain's biggest supermarket chain ..."* |
| 2. *"Nigerian Senate approves $1.5 bln claim on Shell LAGOS - Nigeria's Senate has passed ... "* |
| 3. *"Cocoa farmers issue strike threat. Unions are threatening a general strike in the Ivory Coast ..."* |

| Slice with 139 examples, 3% accuracy. Predictions: 93% "business", labels: 94% "sci/tech" |
|---|
| 1. *"Google Unveils Desktop Search, Takes on Microsoft. Google Inc. rolled out a version of its ..."* |
| 2. *"PalmOne to play with Windows Mobile? Rumors of Treo's using a Microsoft operating system ..."* |
| 3. *"Intel Posts Higher Profit, Sales. Computer-chip maker Intel Corp. said yesterday that earnings ..."* |

Table 3: Representative examples for 2 slices obtained using `InfEmbed` on AGnews. In the 1st slice, the model mis-predicts business text containing country names to be "world". The 2nd slice appears to be mis-labelled, showing `InfEmbed` can detect mis-labeled data.

"science/technology", and most predictions are "business". However looking at the examples, they may in fact be mis-labeled, as they are about businesses and their technology. This shows that `InfEmbed` can also identify *mis-labeled data*.

### 4.5 Detecting Spurious Signals in Boneage Classification

**Overview & Experimental Procedure**: We now artificially inject a signal that spuriously correlates with the label in training data, and see if `InfEmbed-rule` can detect the resulting test errors as well as the spurious correlation. Here, the classification problem is bone-age classification [4] - to predict which of 5 age groups a person is in given their x-ray. Following the experimental protocol of Zhou et al. [2022] and Adebayo et al. [2021], we consider 3 possible spurious signals: tag, stripe, or blur (see Appendix for additional details). For a given signal, we manipulate the bone-age training dataset by injecting the spurious signal into all *training* examples from the "mid-pub" class (short for mid-puberty). Importantly, we leave the test dataset untouched, so that we expect a model trained on the manipulated training data to err on "mid-pub" test examples.

This setup mimics a realistic scenario where x-ray training data comes from a hospital where one class is over-represented, and also contains a spurious signal (i.e. x-rays from the hospital might have the hospital's tag). We then train a resnet50 on the manipulated training data, and apply `InfEmbed-Rule` on the untouched test data to discovered slices with at least 25 examples and at most 40% accuracy (same as for the Imagenet analysis). We repeat this manipulate-train-slice discovery procedure for 3 runs: once for each possible signal.

**Results**: For all 3 runs / signals, `InfEmbed-Rule` returned 2 slices satisfying the rule. From the distribution of labels and predictions, we observe that for each signal, the labels in the most problematic slice are of the "mid-pub" class, showing that `InfEmbed-Rule` is able to detect test errors due to the injected spurious training signal. Furthermore, for each run, around 80% of the slice's opponents are training examples with the spurious training signal, i.e. examples with the "mid-pub" label. Together, this shows `InfEmbed-Rule` can not only detect slices that the model errs on because of a spurious training signal, but also identify training examples with the spurious training signal by using slice opponents.

## 5 Related Work

Error analysis, more generally, has been studied extensively even within the deep learning literature; however, here we focus our discussions on the recently formalized *slice discovery problem*—as termed by Eyuboglu et al. [2022b].

**Error Analysis, Slice Discovery, Related Problems**: To the best of our knowledge, `InfEmbed` is the first SDM based on influence functions. More broadly, it represents the first use of influence functions for clustering and global explainability; they have been used for local explainability, i.e. returning the influence explanation for a *single* test example, but not for clustering, as we do. Pruthi et al. [2020] also define influence embeddings, but use a different definition and also use it for a different purpose - accelerating the retrieval of an example's most influential training examples. Crucially, their embeddings were high-dimensional (exceeding the number of model parameters) and thus ill-suited

---
[4] https://www.kaggle.com/datasets/kmader/rsna-bone-age

for clustering, and lacked any dimension reduction procedure preserving the dot-product property, aside from random projections.

Previous SDMs use a variety of representations as input into a partitioning procedure. Some rely on *external* feature extractors, using PCA applied to pre-trained CLIP embeddings [Eyuboglu et al., 2022b] along with an "error-aware" mixture model, or features from a robustly pre-trained image model [Singla et al., 2021] along with a decision tree. This external reliance limits the generalizability of these SDMs—what if one wanted to perform slice discovery on a domain very different from the one CLIP was trained on, or wanted to consider a new modality, like audio or graphs? Other SDM's use the pre-trained model's last-layer representations [d'Eon et al., 2022], or transform them via SCVIS [Plumb et al., 2022] or UMAP [Sohoni et al., 2020] before applying K-Means clustering. While they are performant, it is less clear *why* this is. In contrast, InfEmbed does not rely on external feature extractors and is theoretically framed by our influence function-based definition of coherency.

Other methods solve problems related to, but not the same as slice discovery. Rajani et al. [2022], Wiles et al. [2022], Hua et al. [2022] leverages the aforementioned SDM's to create *interactive* systems for discovering slices, focusing on improving the user interface and how to explain slices. In contrast, we focus on solving the core slice discovery problem. All these methods use last-layer representations along with K-Means or an "error-aware" mixture model [Hua et al., 2022]. Jain et al. [2022] globally *ranks* examples to identify examples in a *single* failure mode, but being unable to output multiple slices, is not a SDM. Numerous works also rank examples using scores capturing various properties: the degree to which a sample is out-of-distribution [Liu et al., 2020], reliability [Schulam and Saria, 2019], or difficulty Simsek et al. [2022]. Complementary to SDMs are methods for identifying low-performance subgroups when tabular metadata is available [Lakkaraju et al., 2017].

**Influence Functions**: Influence functions [Koh and Liang, 2017] have been used to calculate influences, which can be used for ranking training examples for local explainability [Barshan et al., 2020] for both supervised and unsupervised models [Kong and Chaudhuri, 2021], identifying mis-labeled data interactively [Teso et al., 2021], data augmentation [Lee et al., 2020], active learning [Liu et al., 2021], quantifying reliability [Schulam and Saria, 2019], and identifying mis-labeled data [Pruthi et al., 2020]. However, influence functions have not been used for slice discovery or global explainability.

# 6   Conclusion

We present a method, `InfEmbed`, that addresses the slice discovery problem. Our proposed solution departs from previous work in that it both identifies slices—group of data points— on which a model under-performs, and that satisfy a certain *coherence* property. We formalize coherence—predictions being wrong for the same reasons within a slice—as a key property that all slice discovery methods should satisfy. In particular, we leverage influence functions [Koh and Liang, 2017] to define coherent slices as ones whose examples are influenced similarly by the training data, and then develop a procedure to return coherent slices based on clustering a representation we call influence embeddings. We demonstrate on several benchmarks that `InfEmbed` out-performs current approaches on a variety of data modalities. In addition, we show through case studies that `InfEmbed` is able to recover known errors in a variety of settings, and identifies coherent slices on realistic models.

**Limitations.** One major limitation of the proposed approach is that the validation set needs to include the kind of error that one is seeking to detect. In addition, the proposed procedure suggests to inspect the slice opponents as the root cause of the wrong predictions in a slice. However, it is still up to the user of the proposed Algorithm to infer which feature(s) of the slice opponents is responsible for causing the errors. Despite these limitations, error discovery by clustering influence embeddings can be seen as additional tool in the ML model debugging toolbox.

# Acknowledgments and Disclosure of Funding

Fulton Wang, Sarah Tan, Diego Garcia-Olano and Narine Kokhlikyan were employed by Meta, while Julius Adebayo was employed by Genentech over the course of work on this project. No external funding was received or used by the authors over the course of the project.

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
