# Appendix

## A  Additional Related Work

**Interpretability and Post-hoc Explanations.** Despite initial evidence that explanations might be useful for detecting that a model is reliant on spurious signals [Lapuschkin et al., 2019, Rieger et al., 2020], a different line of work directly counters this evidence. Zimmermann et al. [2021] showed that feature visualizations [Olah et al., 2017] are not more effective than dataset examples at improving a human's understanding of the features that highly activate a DNN's intermediate neuron. Increasing evidence demonstrates that current post hoc explanation approaches might be ineffective for model debugging in practice [Chen et al., 2021, Alqaraawi et al., 2020, Ghassemi et al., 2021, Balagopalan et al., 2022, Poursabzi-Sangdeh et al., 2018, Bolukbasi et al., 2021]. In a promising demonstration, Lapuschkin et al. [2019] apply a clustering procedure to the LRP saliency masks derived from a trained model. In the application, the clusters that emerge are able to separate groups of inputs where, presumably, the model relies on different features for its output decision. This work differs from that in a key way: Lapuschkin et al. [2019] demonstration is to seek understanding of the model behavior and not to perform slice discovery. There is no reason why a low performing cluster should emerge from such clustering procedure.

## B  Practical Low-Rank Influence Function

The main challenge in computing influence is that it is impractical to explicitly form the $H_\theta$ needed to compute influence in $I(z', z) \coloneqq \nabla_\theta L(z'; \theta)^\intercal H_\theta^{-1} \nabla_\theta L(z; \theta)$, unless the model is small, or if one only considers parameters in a few layers. Schioppa et al. [2022] address this problem by forming a low-rank approximation of $H_\theta^{-1}$ without explicitly forming $H_\theta$. Their low-rank implementation first applied the Arnoldi iteration Trefethen and Bau III [1997] to compute an orthonormal basis $(q_1, ..., q_D)$ for the $P$-dimensional Arnoldi subspace $(b, H_\theta b, ..., H_\theta^{P-1})$, as well as the restriction $R$ of $H_\theta$ to the subspace, so that $H_\theta = QRQ^\intercal$, where $Q \coloneqq [q_1, ..., q_P] \in \mathbb{R}^{|\theta| \times P}$ and $R \in \mathbb{R}^{P \times P}$. They choose $P$ to be 200 in their experiments. Crucially, it is computationally feasible to run the Arnoldi iteration even on large models, because it only requires access to $H_\theta$ through Hessian-vector products, not through its explicit formation. Then, leveraging the fact that the Arnoldi subspace of a matrix tends to contain its top eigenvectors, they approximate $R \approx V\lambda V^\intercal$ and $R^{-1} \approx V\lambda^{-1}V^\intercal$, where $V = [v_1, ..., v_D] \in \mathbb{R}^{P \times D}$, $v_1, ..., v_D$ are the top $D$ eigenvectors of $R$, and $\lambda = \text{diag}(\lambda_1, ..., \lambda_D)$, where $\lambda_1, ..., \lambda_D$ are the corresponding eigenvalues. They choose $D$ to be around 50 in their experiments. Crucially, it is computationally feasible to find the top eigenvectors and eigenvalues via eigen-decomposition on an explicitly-formed $R$, because $R$ is of size $P \times P$ with $P = 200$. Finally, they form a low-rank approximation of $H_\theta^{-1}$,

$$\hat{H}_\theta^{-1} \coloneqq M\lambda^{-1}M^\intercal, \text{ where } M = QV \tag{9}$$

and use it to define the *practical influence* of training example $z'$ on test example $z$:

$$\hat{I}(z', z) \coloneqq \nabla_\theta L(z'; \theta)^\intercal \hat{H}_\theta^{-1} \nabla_\theta L(z; \theta). \tag{10}$$

Algorithm B outlines finding the factors needed for $\hat{H}_\theta^{-1}$. Note that for brevity of notation, throughout we will let *configuration* $\Theta \coloneqq (L, f, \theta)$ denote the loss function, model, and parameters.

### B.1  Proof of Lemma 1

Influence embeddings satisfy a critical property - that if two examples have similar influence embeddings, they also tend to have similar influence explanations. This is formalized by the following lemma:

**Lemma 1.** There exists a constant $C$ such that for any test examples $z_i, z_j$, $||E(z_i) - E(z_j)||^2 \leq C||\mu(z_i) - \mu(z_j))||^2$.

---

**Algorithm 3** Finding low-rank factors of Hessian

---

    **procedure** FACTORHESSIAN($\mathbf{Z}'$, $\Theta$, $P$, $D$)
        **Inputs:** training data $\mathbf{Z}'$, configuration $\Theta$, Arnoldi dimension $P$, rank $D$
        $H_\theta \leftarrow \sum_{i=1}^{N'} \nabla_\theta^2 L(z'_i; \theta)$                             ▷ implicitly define HVP
        Run Arnoldi iteration on $H_\theta$ for $P$ iterations to get $Q \in \mathbb{R}^{|\theta| \times P}$, $R \in \mathbb{R}^{P \times P}$
        $V, \lambda \leftarrow$ top-$D$ eigenvectors / values of $R$ via SVD
        $M \leftarrow QV$
        **Return:** $M, \lambda$
    **end procedure**

---

*Proof.* The proof follows from the Cauchy-Schwartz inequality, and the fact that influence is the dot-product of influence embeddings.

$$
\begin{aligned}
||E(z_i) - E(z_j)||^2 &= \sum_{n=1}^{N'} (\hat{I}(z'_n, z_i) - \hat{I}(z'_n, z_j))^2 \\
&= \sum_{n=1}^{N'} (\mu(z'_n)^\intercal \mu(z_i) - \mu(z'_n)^\intercal \mu(z_j))^2 \\
&= \sum_{n=1}^{N'} (\mu(z'_n)^\intercal (\mu(z_i) - \mu(z_j)))^2 \\
&\leq \sum_{n=1}^{N'} ||\mu(z'_n)||^2 ||\mu(z_i) - \mu(z_j)||^2 \\
&\leq C ||\mu(z_i) - \mu(z_j)||^2, \text{ where}
\end{aligned}
$$

$C := \sum_{n=1}^{N'} ||\mu(z'_n)||^2$ does not depend on $i$ nor $j$.     □

### B.2 Description of `InfEmbed-Rule`

Note that the key hyperparameter of the InfEmbed method is $K$, the number of slices to return. In practice, it may not be intuitive for a user to choose $K$. Instead, the user may want to know if there exists any coherent slices that are problematic, as defined by satisfying a rule: has accuracy less than some threshold, and number of examples above some threshold. Therefore, we also propose a procedure that recursively clusters influence embeddings until slices satisfying the rule are found, or until the slices are too small. The approach is analogous to building a tree to identify slices satisfying the rule, where the splits are determined by K-Means clustering of influence embeddings. In addition to letting the user specify more intuitive hyperparameters, this procedure also has the advantage that if a large slice with sufficiently low accuracy is found, it will not be clustered further. Algorithm 1 outlines this `InfEmbed-Rule` method. Its inputs are the same as `InfEmbed`, except instead of specifying $K$, the number of slices, one specifies accuracy threshold $A$, size threshold $S$, and branching factor $B$, which specifies how many clusters the K-means call in each step of the recursion should return. Its outputs is a set of slices, each having accuracy less than $A$ and size greater than $S$. In practice, the recursion proceeds to a maximum depth.

## C DcBench

**Overview.** Eyuboglu et al. [2022b] formalized the slice discovery problem and introduced Dcbench [Eyuboglu et al., 2022a], a benchmark, consisting of pre-trained models across a variety of data sets for testing any new SDM. For each dataset in the benchmark, a manipulation is applied to the dataset in order to induced one of three kinds of errors in a collection of samples—slice—, and a model is trained on the modified dataset to exhibit the injected error. To assess a new SDM, the partitioning returned by the method is then compared to the ground-truth slices used to generate the model. The collection of datasets in dcbench includes natural images (ImageNet, and CelebA), Medical images based on the MIMIC Chest X-Ray, and Medical Time-Series Data of electroencephalography (EEG) signals. A collection of 1235 models were trained for various dataset, model, and error type groups. In addition to the benchmark, Eyuboglu et al. [2022b] introduced Domino, an SDM, that uses a mixture model that models the generative process of a slice. The input to the mixture model is a multi-model embedding (CLIP or ConVIRT) of the test data, and the mixture model assumes that the embedding, label, and predictions are independent conditioned on the slice. Eyuboglu et al. [2022b] demonstrate that Domino outperforms or matches approaches based

---

**Algorithm 4** Procedure finding slices with low accuracy and large size

---

**procedure** RULEFIND($\boldsymbol{\mu}, A, S, B$)
    **Inputs:** $\boldsymbol{\mu}$ a list of influence embeddings, accuracy threshold $A$, size threshold $S$, branching factor $B$
    **Outputs:** a set of lists of influence embeddings. Each set of influence embeddings corresponds to a slice with accuracy $< A$ and size $> S$
    $acc \leftarrow$ accuracy of examples represented in $\{\mu_i\}$
    $size \leftarrow$ number of examples represented in $\{\mu_i\}$
    **if** $acc \leq A$ and $size \geq S$ **then**
        **Return:** $\{\boldsymbol{\mu}\}$
    **end if**
    **if** $size < S$ **then**
        **Return:** $\{\}$
    **end if**
    $\boldsymbol{r} \leftarrow$ K-Means($\boldsymbol{\mu}, B$)                     ▷ get cluster assignments
    $\boldsymbol{r}_k \leftarrow \{\mu_i\}_{i:r_i=k}$ for $k \in [B]$           ▷ embeddings for each cluster
    $F = \{\}$
    **for** $k \in [B]$ **do**                       ▷ search within each cluster
        $F \leftarrow F \cup \{\text{RuleFind}(\{\mu_i\}_{i:r_i=b}, A, S, B)\}$
    **end for**
    **Return:** $F$
**end procedure**
**procedure** INFEMBED-RULE($A, S, B, \mathbf{Z}, \mathbf{Z}', \Theta, P, D$)
    $\boldsymbol{\mu} \leftarrow$ GetEmbeddings($\mathbf{Z}, \mathbf{Z}', \Theta, P, D$)
    $F \leftarrow$ RuleFind($\boldsymbol{\mu}, A, S, B$)       ▷ a set of lists of embeddings, with each embedding corresponding to a test example
    **Return:** the partition of test dataset induced by $F$
**end procedure**

---

on last-layer or vision-transformer-based embeddings. Consequently, we compare against Domino on dcbench since it is, as of the writing of this paper, the state-of-the-art.

**Experimental Procedure.** The dcbench benchmark consists of a collection of slice discovery tasks. Each task includes a series of artifacts: the model, base dataset, model activations, validation and test predictions, clip activations for the test and validation data, and ground-truth slices. Domino was evaluated using two kinds of models: a synthetic and trained model. We restrict our focus to the trained models, only, since our proposed approach only applies to trained models. We compute the precision-at-10 metric, similar to domino, across all data modalities and error type.

**Results.** We present a comparison of the Precision-at-10 metric between Domino and the influence embeddings approach in Table 1. The influence embedding approach outperforms domino across all settings except on the Noisy Label task for the EEG data modality.

## C.1 Datasets

All of the dcbench dataset, models, and task information is publicly available via the opensource repository: https://github.com/data-centric-ai/dcbench

# D SpotCheck

**Model**: Following Plumb et al. [2022], we train a ResNet-18 model for each dataset and blindspot specification.

## D.1 Datasets

The Spotcheck benchmark is publicly described in the paper by Plumb et al. [2022]. The SpotCheck benchmark (Plumb et al., 2022) is based on a synthetic task consisting of 3 semantic features that can be easily con- trolled to determine the number of 'blindspots' in a dataset. A blindspot is feature

responsible for a model's mistake. Similar to dcbench, a model trained on data generated from SpotCheck is induced to make mistakes on an input that has a set of blindspots. The overall task is to predict the presence of a square in the image. Attributes of the input image such as the background, object color, and co- occurrence can be varied to induce mistakes. Plumb et al. (2022) find that current SDM approaches struggle in the presence of several blindspots. Consequently, they pro- pose PlaneSpot, an SDM that uses the representation of the model's last layer, projected to 2 dimensions, along with the model's 'confidence' as part of a mixture model to partition the dataset. They show that PlaneSpot outperforms current approaches as measured by discovery and false discovery rates. We replicate the SpotCheck benchmark (for 4 variable features) and compare InfEmbed to PlaneSpot.

# E   Imagenet & High Dimensional Settings

**Overview & Experimental Procedure:** The Imagenet validation data Deng et al. [2009] contains 1000 classes, with 50 examples per class. We use the `InfEmbed-Rule` procedure to find slices with at most 40% accuracy, and at least 25 examples.

We use a Resnet18 trained on Imagenet, provided in the torchvision PyTorch library. The model achieves 69.8% accuracy. To compute influence embeddings, we consider gradients in the "fc" and "layer4" layers, which contain a total of 8.9M parameters. As in all experiments, we compute influence embeddings of $D = 100$ dimensions using an Arnoldi dimension of $P = 500$. To explain the root-cause of predictions in a given slice, we compute the strongest opponents of the slice. These are the training examples whose influence on the *slice*, i.e. influence on the *total* loss over all examples in the slice is the most negative, i.e. most harmful. Due to the properties of influence embeddings, the influence of a training example on a slice is simply the dot-product between the influence embedding of the training example, and the *sum* of the influence embeddings in the slice. For simplicity, we search for opponents within the test dataset, i.e. treating the test data as the training data. Since there is no distribution shift in Imagenet, for explanatory purposes this is an acceptable approximation.

**Results:** Applying the above rule, we find 25 slices, comprising a total of 954 examples, whose overall accuracy is 34.9%. Figure E displays 4 of the slices, where each row contains the distribution of predicted and true labels, the 4 examples nearest the slice center in influence embedding space, and the 4 strongest opponents of the slice. First, we see that the slices have the label homogeneity property as explained in Section 3.5 - the predicted and true labels typically come from a small number of classes. In the first slice (counting from the top), the true label is mostly "sidewinder", and often predicted to be "horned viper". The slice's strongest opponents are horned vipers, hard examples of another class, which the model likely models similarly to sidewinders. Thus the presence of the former drives the prediction of the latter towards horned vipers and away from sidewinders, increasing loss. A similar story holds for the second slice, where breastplates are often predicted to be cuirasses. Interestingly, the first opponent may actually be mis-labeled (cuirasses are breastplates fused with backplates, which it may lack), reflecting the fine line between hard and mis-labeled opponents examples. The third and fourth slices are interesting in that each slice is predominantly of *two* labels. For example, in the third slice, border collies are strong opponents for both borzois and collies (which are a different dog breed than border collies), causing them to be mis-predicted as border collies. This makes sense given that all three dog breeds look similar. In the fourth slice, vine snakes and analog clocks are often mis-predicted. A priori, we would not know that predictions for these seemingly unrelated classes would be wrong for the same reasons. However, looking at the opponents, we see all of them have a clock hand, which turns out to be similar to a snake. Also, because the opponents are of a variety of classes, the mis-predictions are also of a variety of classes.

Finally, we also list a few other discovered slices: 1) a slice where "gown" (wedding dress) is mis-predicted to be "groom", whose opponents are images labeled "groom" containing both a groom and spuriously-correlated gown, 2) a slice where "windsor tie" is mis-predicted to be suit, due to similar spurious correlations, 3) a slice where mis-predicted "sunglasses" are due to examples which contain sunglasses, but are labeled as other present classes, like "lipstick" and "bib", 4) two different classes (i.e. index) actually refer to the same object - "maillot".

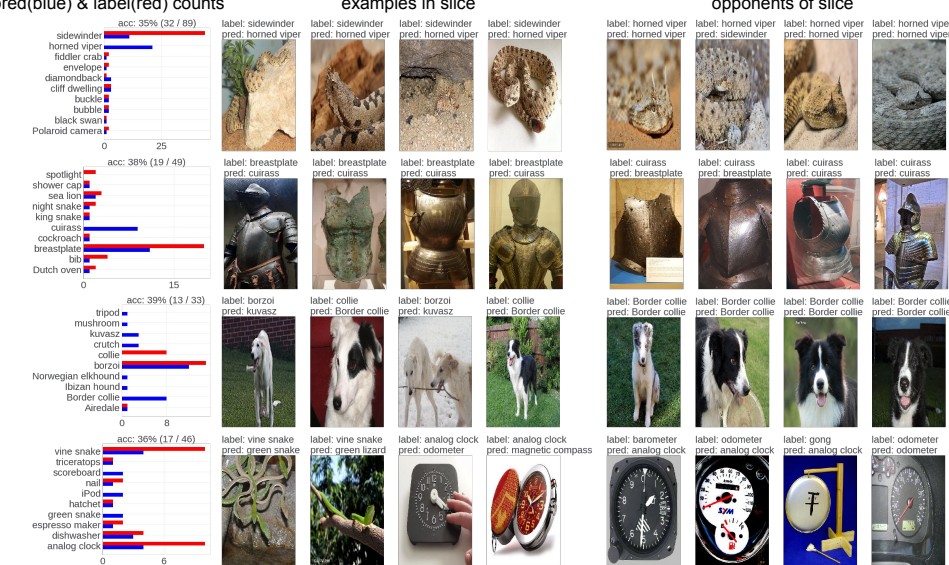

Figure 3: For slices discovered by applying `InfEmbed-Rule` to Imagenet, we show the distributions of labels (red) and predictions (blue) in the slice (left), examples from the slice (middle), and opponents of the slice (right).

## F   AGNews & Mis-labeled Data

**Overview & Experimental Procedure:** The AGnews test data Zhang et al. [2015] contains 4 classes (Business, Sci/Tech, Sports, World) with 1900 examples per class. We use a BERT base model fine-tuned on the training set [5], which achieves 93.75% accuracy on the test data which results in 475 test errors. We use the InfEmbed method of Algorithm 1 with $K = 25$ to find slices with at most 10% accuracy, and at least 10 examples since the total number of errors is small.

To compute influence embeddings, we consider gradients in the "bert.pooler.dense" and "classifier" layers, which are the top 2 linear layers of the model and contain a total of 590K parameters parameters. As in all experiments, we compute influence embeddings of $D = 100$ dimensions using an Arnoldi dimension of $P = 500$.

**Results:** We find 9 slices that account for 92% of errors (438 out of 475). Some interesting patterns we observe:

- Sci/Tech examples predicted as Business (30% ) & Business predicted as Sci/Tech (25%) are 55% of errors.
- Business examples predicted as World (10.3%) & World predicted as Business (9.4%) are 19.7% of errors.
- Sci/Tech examples predicted as World (8.5%) & World predicted as Sci/Tech (7.3%) are 15.8% of errors.
- Sports articles are least likely to get predicted for other genres and only a few Business and Sci/Tech examples get predicted as Sports ( 4.1% and 2.7% respectively of errors. )

Table 4 displays representative examples (ie, those closet to the cluster centers ) from 3 of the slices. For Sci/Tech news articles that are predicted as Business articles, we see Google, PalmOne and Intel all referenced, but with references to investment bank reports, Stock symbols and earning reports, the true class is a bit ambiguous and illustrates why the model has the most troubles distinguishing between the two ( they account for 55% of errors ). In fact, it is possible examples in this slice are actually mis-labeled, showing that `InfEmbed` can be used to detect systematically mis-labeled data. For Business articles that are predicted to be World articles we see references to the grocer Tesco,

---

[5]https://huggingface.co/fabriceyhc/bert-base-uncased-ag_news

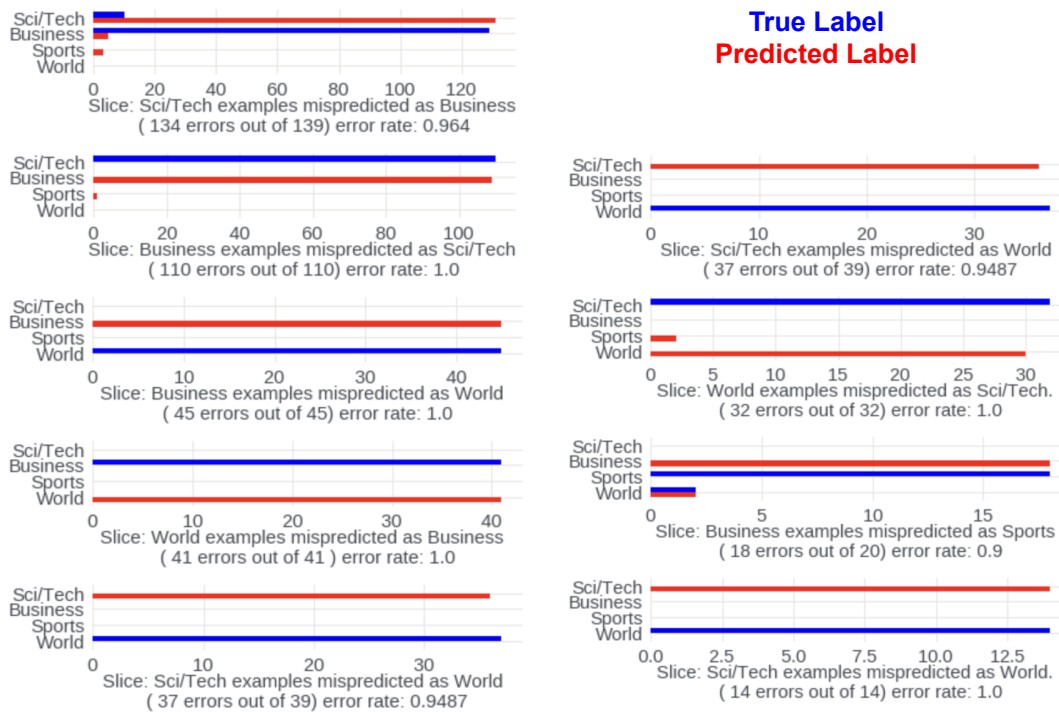

Figure 4: Predicted vs True label distributions for 9 slices found for AGnews data

Shell and cocoa farmers, but with reference to Britain, the Nigerian Senate and protests in the Ivory coast which again makes their class a bit ambiguous; accounting for 20% of model errors. Finally, for Business examples that are predicted as Sports, which is one of the smaller error slices we find, we see less ambiguous error instances which the model has trouble with.

Furthermore, we display in Figure 4 the distribution of labels and predictions for the 9 slices, as well as their error rate. We see that the slices possess the label homogeniety property.

## G   Detecting Spurious Signals in Boneage Classification

**Bone Age Dataset**: We consider the high stakes task of predicting the bone age category from a radiograph to one of five classes based on age: Infancy/Toddler, Pre-Puberty, Early/MiD Puberty, Late Puberty, and Post Puberty. This task is one that is routinely performed by radiologists and as been previously studied with a variety of DNN. The dataset we use is derived from the Pediatric Bone Age Machine learning challenge conducted by the radiological society of North America in 2017 **?**. The dataset consists of 12282 training, 1425 validation, and 200 test samples. We resize all images to (299 by 299) grayscale images for model training. We note here that the training, validation, and test set splits correspond to similar splits used for the competition, so we retain this split.

**Model and Hyperparameters**: We consider and a Resnet-50 model. The small DNN consists of: conv-relu-batchnorm-maxpooling operation successively, and two fully connected layers at the end. All convolutional kernels have stride 1, and kernel size 5. We train this model with SGD with momentum (set to 0.9) and an initial learning rate of 0.01. We use a learning rate scheduler that decays the learning rate every 10 epochs by $\gamma = 0.1$.

**Overview & Experimental Procedure**: We now artificially inject a signal that spuriously correlates with the label in training data, and see if InfEmbed-rule can detect the resulting test errors as well as the spurious correlation. Here, the classification problem is bone-age classification - to predict which of 5 age groups a person is in given their x-ray. Following the experimental protocol of Zhou et al. [2022] and Adebayo et al. [2021], we consider 3 possible signals that can be added to an x-ray. Importantly, spurious correlations involving these signal have been difficult to detect in past work. For a given signal, we manipulate the bone-age training dataset by injecting the spurious

Examples from slice with 139 examples, 3% accuracy. Predictions are 93% "business", labels are 94% "sci/tech"

| |
|---|
| 1. *"Google Unveils Desktop Search, Takes on Microsoft Google Inc. (GOOG.O: Quote, Profile, Research) on Thursday rolled out a preliminary version of its new desktop search tool, making the first move against ..."* |
| 2. *"PalmOne to play with Windows Mobile? Rumors of Treo's using a Microsoft operating system have been circulating for more than three years. Now an investment bank reports that PalmOne will use a ..."* |
| 3. *"Intel Posts Higher Profit, Sales Computer-chip maker Intel Corp. said yesterday that earnings for its third quarter were $1.9 billion – up 15 percent from the same quarter a year ago ..."* |

Examples from slice with 45 examples, 0% accuracy. Predictions are 100% "world", labels are 100% "business"

| |
|---|
| 1. *"British grocer Tesco sees group sales rise 12.0-percent (AFP) AFP - Tesco, Britain's biggest supermarket chain, said that group sales grew by 12.2 percent in the third quarter, driven by strong performances from its stores ..."* |
| 2. *"Nigerian Senate approves $1.5 bln claim on Shell LAGOS - Nigeria's Senate has passed a resolution asking Shell's Nigerian unit to pay $1.5 billion in compensation to oilfield communities for pollution, a Senate spokesman said. "* |
| 3. *"Cocoa farmers issue strike threat. Unions are threatening a general strike in the Ivory Coast in a protest against the prices farmers are paid for their cocoa supplies."* |

Examples from slice with 20 examples, 10% accuracy. Predictions are 90% "world", labels are 90% "business"

| |
|---|
| 1. *"Perry OKs money for APS as more accusations arise. The state's Adult Protective Services agency will get an emergency infusion of $10 million to correct the kinds of problems that have arisen in El Paso."* |
| 2. *Sign off, then sign in. G. Michael Caggiano Jr. lies awake at night thinking about bank signs. He ponders them during breakfast, while brushing his teeth, and quot;constantly quot; during the day, he says."* |
| 3. *"Stanley set sights on Elland Road for casino Stanley Leisure plc has announced a Stanley Casinos Limited plan to develop a casino complex on land adjacent to Leeds United's Elland Road stadium."* |

Table 4: Representative test examples for 3 high error slices obtained using `InfEmbed` on AGnews

signal into all *training* examples from the "mid-pub" class (short for mid-puberty, one of the 5 age groups). Importantly, we leave the test dataset untouched, so that we expect a model trained on the manipulated training data to err on "mid-pub" test examples; the spurious signal the model associated with "mid-pub" is missing, so that the model, having not associated other features with "mid-pub", finds no evidence to deem it "mid-pub". This setup mimics a realistic scenario where x-ray training data comes from a hospital where one class is over-represented, and also contains a spurious signal (i.e. x-rays from the hospital might have the hospital's tag).

We then train a resnet50 on the manipulated training data, and apply `InfEmbed-Rule` on the untouched test data to discovered slices with at least 25 examples and at most 40% accuracy (same as for the Imagenet analysis). To assess whether `InfEmbed-Rule` succeeded at detecting the model's reliance on the spurious training signal, we 1) take the most problematic slice, i.e. discovered slice with the lowest accuracy, and see if its labels are mostly "mid-pub" - the class we know a priori to be under-performing in the test dataset due to spurious training signal injection, and 2) examine whether the slice opponents are training examples with the spurious training signal, i.e. of the "mid-pub" class. This lets us confirm whether the errors in the slice are due to the spurious training signal (as opposed to other root-causes), and whether using `InfEmbed-Rule` along with slice opponents analysis lets us discover the spurious training signal. We repeat this manipulate-train-slice discovery procedure for 3 runs: once for each possible signal.

**Results**: For all 3 runs / signals, `InfEmbed-Rule` returned 2 slices satisfying the rule. In Figure **??**, for each signal, we show the distribution of labels and predictions for the most problematic slice

(left column) and the same distribution for that slice's top-50 opponents (right column). We see that for each signal, the labels in the most problematic slice are of the "mid-pub" class, showing that `InfEmbed-Rule` is able to detect test errors due to the injected spurious training signal. Furthermore, for each run, around 80% of the slice's opponents are training examples with the spurious training signal, i.e. examples with the "mid-pub" label. Together, this shows `InfEmbed-Rule` can not only detect slices that the model errs on because of a spurious training signal, but also identify the root-cause of those errors by using slice opponents to identify training examples with the spurious signal.

# H    Applying Influence Embeddings to Limited Data Settings & Using Attributes

We use the COVID-19 Chest X-Ray Dataset[6]. This dataset, provided in the torchxrayvision PyTorch library, has 535 images of chest X-rays. We use the COVID-19 label, a binary label, where there are 342 cases of COVID. We leverage a pretrained DenseNet model (densenet121-res224-all) provided in the torchxrayvision library, trained to predict multiple diseases, and adapt it to predict the COVID label by replacing the last fully-connected layer of the model. We then fine-tune the model, including this last layer, on a training split of the COVID dataset using a binary cross-entropy loss. We follow standard normalization techniques for these datasets, including using a central crop of 224x224 pixels and normalizing the image values in the $[-1024, 1024]$ range. The resulting model achieves an accuracy of 76.6% on the test set of 107 points. We apply InfEmbed, taking gradients with respect to the last fully connected layer of the model. We then apply K-Means clustering with $k = 3$.

**Results.** The first cluster had an accuracy of 85%, capturing the majority of correctly classified samples. Besides correctly classified positive and negative samples, the first cluster also had five false positives and nine false negatives. The second cluster consisted of four samples that were false positives and two samples that were true negatives, while the third cluster had six samples that were false positives, and one sample that was a false negative. To a layperson not trained in interpreting radiology images, it may be hard to pick out differences between images in the three clusters. To study the differences between the three clusters, we inspected attributes and metadata provided in the dataset.

Besides the COVID-19 label, the dataset had 18 additional labels for various diseases and infections, from Aspergillosis to Varicella. The full list can be found in Table 5; see Cohen et al. [2020] for more details. We computed the incidence of these diseases for each of the true negatives, true positives, false negatives, false positives samples by cluster. Table 5 presents the results. We observe that false positive samples in all three clusters tended to have pneumonia and tuberculosis, but not COVID-19. Looking at true positives, all the samples in cluster 1 that had COVID-19 also had pneumonia. However, samples with pneumonia do not always have COVID-19; rather, sometimes they have other lung diseases such as tuberculosis, SARS, etc.

A differentiator between the three clusters is the incidence of additional diseases. Some false positive samples in cluster 2, in addition to pneumonia and tuberculosis, also had legionella, while those in cluster 3 in addition had SARS or pneumocystis. The false positive samples in cluster 1 had more diseases, such as Herpes, Klebsiella, and several others. Many of these diseases are lung diseases; these findings illustrate that the model may be having a harder time differentiating between different lung diseases and COVID-19.

---

[6]https://github.com/ieee8023/covid-chestxray-dataset

| | True Negatives | | | False Positives | | | False Negatives | | | True Positives | | |
| --- | --- | --- | --- | --- | --- | --- | --- | --- | --- | --- | --- | --- |
| | C1 | C2 | C3 | C1 | C2 | C3 | C1 | C2 | C3 | C1 | C2 | C3 |
| Aspergillosis | 0.04 | 0.0 | NaN | 0.0 | 0.00 | 0.00 | 0.0 | NaN | 0.0 | 0.0 | NaN | NaN |
| COVID-19 | 0.00 | 0.0 | NaN | 0.0 | 0.00 | 0.00 | 1.0 | NaN | 1.0 | 1.0 | NaN | NaN |
| Chlamydophila | 0.00 | 0.0 | NaN | 0.0 | 0.00 | 0.00 | 0.0 | NaN | 0.0 | 0.0 | NaN | NaN |
| H1N1 | 0.00 | 0.0 | NaN | 0.0 | 0.00 | 0.00 | 0.0 | NaN | 0.0 | 0.0 | NaN | NaN |
| Herpes | 0.00 | 0.0 | NaN | 0.2 | 0.00 | 0.00 | 0.0 | NaN | 0.0 | 0.0 | NaN | NaN |
| Influenza | 0.00 | 0.0 | NaN | 0.0 | 0.00 | 0.00 | 0.0 | NaN | 0.0 | 0.0 | NaN | NaN |
| Klebsiella | 0.12 | 0.0 | NaN | 0.2 | 0.00 | 0.00 | 0.0 | NaN | 0.0 | 0.0 | NaN | NaN |
| Legionella | 0.00 | 0.5 | NaN | 0.0 | 0.25 | 0.00 | 0.0 | NaN | 0.0 | 0.0 | NaN | NaN |
| MERS-CoV | 0.00 | 0.0 | NaN | 0.2 | 0.00 | 0.00 | 0.0 | NaN | 0.0 | 0.0 | NaN | NaN |
| MRSA | 0.00 | 0.0 | NaN | 0.0 | 0.00 | 0.00 | 0.0 | NaN | 0.0 | 0.0 | NaN | NaN |
| Mycoplasma | 0.00 | 0.0 | NaN | 0.0 | 0.00 | 0.00 | 0.0 | NaN | 0.0 | 0.0 | NaN | NaN |
| Nocardia | 0.04 | 0.0 | NaN | 0.0 | 0.00 | 0.00 | 0.0 | NaN | 0.0 | 0.0 | NaN | NaN |
| Pneumocystis | 0.16 | 0.0 | NaN | 0.2 | 0.00 | 0.33 | 0.0 | NaN | 0.0 | 0.0 | NaN | NaN |
| Pneumonia | 0.92 | 1.0 | NaN | 1.0 | 0.25 | 1.00 | 1.0 | NaN | 1.0 | 1.0 | NaN | NaN |
| SARS | 0.12 | 0.0 | NaN | 0.0 | 0.00 | 0.50 | 0.0 | NaN | 0.0 | 0.0 | NaN | NaN |
| Staphylococcus | 0.00 | 0.0 | NaN | 0.0 | 0.00 | 0.00 | 0.0 | NaN | 0.0 | 0.0 | NaN | NaN |
| Streptococcus | 0.04 | 0.0 | NaN | 0.0 | 0.00 | 0.00 | 0.0 | NaN | 0.0 | 0.0 | NaN | NaN |
| Tuberculosis | 0.08 | 0.0 | NaN | 0.0 | 0.25 | 0.00 | 0.0 | NaN | 0.0 | 0.0 | NaN | NaN |
| Varicella | 0.04 | 0.0 | NaN | 0.0 | 0.00 | 0.00 | 0.0 | NaN | 0.0 | 0.0 | NaN | NaN |

Table 5: Average characterization of the true negatives (tn), false positives (fp), false negatives (fn), and true positives (tp) of each of the 3 clusters C0, C1, and C2. We compute the characterization by averaging the 20-dimensional ground-truth label of each sample (that describes the pathologies associated with that sample) across all samples belonging in that group. For groups where no sample is assigned, the characterization is NaN.