# OpenReview forum: "Error Discovery By Clustering Influence Embeddings"
_NeurIPS.cc/2023/Conference — NeurIPS 2023 poster_

### Official Review · Reviewer_yzdi · 2023-06-27

**Soundness:** 3 good
**Presentation:** 3 good
**Contribution:** 3 good
**Rating:** 7
**Confidence:** 4

**Summary:**

This work presents a method for discovering subsets of the test set of a multi-class classification task on which a trained model incorrectly classifies a large portion due to the same root cause. The method uses a factorized low-rank approximation of a bilinear influence function, parameterized using the Hessian of loss function with respect to the parameters of the classifier, to project high dimensional influence explanations into lower dimensional representations of test examples, naturally called influence embeddings. The influence embeddings preserve structure of high dimensional influence explanation space and are used to cluster the test examples into subsets using k-means. The influence embeddings are also shown to have purpose in finding slices of the test set that satisfy certain size and failure properties and can be used to find the most problematic training examples causing the failures. Empirical results show that the proposed method outperforms other slice discovery methods on benchmarks and is able to find known errors in several settings.

**Strengths:**

* The problem is important and well-motivated
* The general method is easy to understand and seems scalable
* Empirical evidence is strong and shows the method is very suitable for the specified problem
* The paper as a whole is fairly well-written

**Weaknesses:**

* Given the work of Schioppa et al. [2022], the methodological contribution seems quite small and straight-forward
* The clustering algorithms and procedures using clustering as a subroutine (InfEmbed-Rule) seem to treat the clustering algorithm as an after-thought. There is no justification for why k-means is used.
* The theoretical results are slightly over-stated in the introduction. It seems like more can be done on the theoretical side to motivate the method. Lemma 1 and Section 3.5 contain interesting results that could be strengthened and elaborated to improve the justification for the presented method.
* The writing in the background section could be improved a bit. It seems like notation is introduced in an unnatural order and in a slightly imprecise way. Also the definition of the Hessian (line 98) is unclear: which examples are used to define the Hessian? Training examples? Test examples? Or maybe both?
* Figure 1 really is not that helpful in understanding the method and takes up quite a bit of space.

**Questions:**

* Why is k-means the clustering algorithm of choice? It seems like specifying the number of slices could hinder the methods effectiveness? DBSCAN was used in an ablation and seems like a better (maybe not the best) choice since it is density based.
* Why is a hierarchical clustering (tree or DAG shaped clustering) not used for InfEmbed-Rule? Why do we need to re-cluster using k-means to find slices with user defined properties?
* How is the subset of the training set passed to FactorHessian chosen? (Section 4, lines 229-230)
* How would this slice discovery method be used to improve the model?
* How does the quality of the low-rank approximation of the influence function affect the quality of slices discovered?


**Limitations:**

The choice of influence function seems to be chosen by fiat and not really justified. The entire method hinges on the specified influence function being the best. It does seem to be used in the sub-community, but is not justified by the authors. I think the limitations of the method with respect to the issues with the choice of influence function should be addressed.

Is it possible that this method could be used to cause harm in some way. Could the authors address this with respect to the method presented?

---

> ### Author Rebuttal · Authors · 2023-08-10
>
> Thank you for your detailed feedback. We address your concerns below.
>
> **Weaknesses**
> - **Methodological Contribution**: We agree that the actual computation of the influence embeddings being clustered in the main algorithm is builds on the work of Schioppa et al. [2022]. However, we make 3 key contributions:
> 1. Our work is the first to connect one popular problem in model debugging (slice discovery), with a popular tool in model debugging (influence functions).
> 2. We theoretically derive what the representation to cluster in a slice discovery algorithm should be in terms of an influence function-based definition of coherency. This is significant, given that past slice discovery algorithms choose to cluster various representations (CLIP embeddings, last-layer representations etc) without theoretically explaining those representations should enable effective slice discovery.
> 3. We are the first to use influence functions (of any kind, including but not limited to Schioppa et al) for global explainability (simultaneously explaining predictions for the entire test dataset), unlike past work, which solely focused on local explainability (explaining one prediction at a time, independent of other predictions).
>
> - **Why K-Means**: Our core contribution is identifying what representation should be clustered for slice discovery, and not how to cluster those representations. Furthermore, we tested 3 clustering algorithms (DBSCAN, spectral clustering, gaussian mixture model) on the Spotcheck benchmark, and found that 1) performance was not sensitive to the choice of clustering algorithm and 2) no matter what clustering algorithm was used on influence embeddings, the performance beat the Domino and PlaneSpot baselines.  These results were unfortunately hidden in Appendix J, and will be moved to the main body.
>
> - **Theoretical Results**:  Thank you for the suggestion. We will rewrite the presentation of the theoretical results. The slice discovery field has to-date been without much theoretical underpinning, and this submission hopes to be a first step towards addressing that gap. In this work, we focus on showing intuition for why simply using K-Means to cluster influence embeddings implicitly leads to desirable properties that other slice discovery methods needed to explicitly encourage, suggesting that influence embeddings are the right representation to cluster.
> - **Background Section**: Thank you for the writing feedback; we will rewrite to clarify that training examples are used to define the Hessian.
> - **Figure 1**: Thank you for the feedback, we will update the figure to make the steps more clear.
>
> **Questions**
> - **Why K-Means (and not DBSCAN)?**: We definitely agree using DBSCAN and not having to choose the K, the number of slices, beforehand is a good idea that would make InfEmbed easier to use, especially since the various clustering algorithms performed similarly (Appendix J results).  We will highlight this possibility in the main body.
> - **Hierarchical Clustering**:  InfEmbed-Rule can actually be viewed as a form of top-down hierarchical clustering (using K-means to subdivide existing clusters) with 1 important modification - we do not further cluster a cluster if it already has sufficiently low accuracy and large size. The reason for doing so is that for error analysis, we want to find the largest cluster with sufficiently low accuracy.
> - **Factor Hessian**: We chose an uniformly random subset. We will add the detailed numbers in the Appendix.
> - **How would this slice discovery method be used to improve the model**: This is a great question. Once we identify a high-error slice, we can identify the slice opponents - the examples whose inclusion in the training data increases the loss on the slice (see Section 3.6), remove them, and either retrain or fine-tune the model.  If the examples in a high-error slice turn out to be mis-labeled (as was the case for some slices in the AGNews case study of Section 4.4), we can identify their proponents (training examples which supported the prediction for the wrong label) and remove them. Model improvement is a particularly important challenge, and interesting future work.
> - **Quality of low-rank approximation**: The quality of the low-rank approximation can indeed affect the quality of the discovered slices.  We found that a small Arnoldi dimension, rank, or size of training data subset used for the Hessian could lead to worse quantitative and qualitative results. However, we also found that following the recommendations by Schioppa et al. [2022] led to satisfactory results.
>
> **Limitations**
> - **Choice of influence function**: We chose to use influence functions because 1) they have been shown to be powerful model debugging tools (Koh and Liang, 2017, Han et al 2020, Kong et al 2020), 2) they provide actionable next steps for model improvement (remove opponents of examples in high-error slices and retrain), and 3) using them leads us to derive influence embeddings, which possess appealing properties when clustered. We will provide additional justification in the draft.
> - **Harms**: As with every model diagnosis or understanding method, there is a risk of the method giving the practitioner a false sense of safety. If the diagnosis is wrong, and an action is taken based on the wrong diagnosis, this may have unintended consequences.
>
> We thank you for the feedback, and would be happy to address any other questions that you have. We encourage you to reconsider your score in light of our response.
>
> **References**\
> [Koh and Liang, 2017]: Understanding black-box predictions using influence functions.  ICML 2017.\
> [Han et al, 2020]: Explaining black-box prediction and unveiling data artifacts through influence functions.  ACL 2020.\
> [Kong et al, 2020]: Understanding instance-based interpretability of variational autoencoders.  Neurips 2020.

---

> > ### Comment · Reviewer_yzdi · 2023-08-14
> >
> > Thank you for your detailed response. Given the promised changes and responses, I am happy to update my score above.

---

### Official Review · Reviewer_6oL7 · 2023-07-03

**Soundness:** 3 good
**Presentation:** 3 good
**Contribution:** 3 good
**Rating:** 7
**Confidence:** 3

**Summary:**

The paper presents a method to discover groups of test examples on which the model performs badly, and the misclassification of the examples is caused by the same reason (defined as coherence).
This problem is known as slice discovery. The method leverages influence functions to compute the influence explanation for each test example. This generates a vector of the influence of each training example on the test example. Given that this vector is high-dimensional, they derive influence embeddings that are used to cluster them by applying K-Means. The authors propose a variant in which there is no need to specify the number of clusters but only the minimum size of the slice and the maximum accuracy. The method is evaluated on several datasets.

**Strengths:**

- the paper is well written and structured clearly. The contributions are clearly reported, and the related works are described.
- the problem is relevant because finding the training examples that influence a group of test examples on which the model underperforms is the first step to debug a model and to fix sporious correlations in the data.
- using influence functions to address this problem is novel according to the provided references and interesting
- the author formalizes the coherence desideratum
- the metho is evaluated extensively on multiple datasets (text and images) and considering multiple types of causes that induce the model to make a classification error

**Weaknesses:**

_Reproducibility_: it is not clear if the source code of the experiments will be made available upon acceptance. The datasets are publicly available.

**Questions:**

- how has been selected the number of cluster K for each experiment with InfEmbed?
- I couldn't find the value of the hyperparameter P, D and K on dcbench and SpotCheck

**Limitations:**

A very brief discussion of the limitations is addressed in the supplementary material. It would be useful for the reader to have the limitation section in the main text, even if there is limited space for this addition.

---

> ### Author Rebuttal · Authors · 2023-08-10
>
> Thank you for your detailed feedback. We address your concerns below.
>
> **Weaknesses**
> - **Reproducibility**: Regarding reproducibility, we plan to release all our code, datasets, and additional artifacts to replicate our analyses.
>
> **Questions**
> - **Choosing number of clusters K for InfEmbed experiments**: For K (number of clusters), we follow past work to ensure comparability of quantitative results: when comparing to Domino in Table 1, we follow Domino and use K=25, and when comparing to various methods on the SpotCheck benchmark in Table 2, we set the number of clusters K using the Bayesian information criterion (BIC), following how the SpotCheck benchmark authors chose K for their slice discovery method, PlaneSpot
> - **Hyperparameters**: For K (number of slices), we follow past work to ensure comparability of quantitative results: when comparing to Domino in Table 1, we follow Domino and use K=25, and when comparing to various methods on the SpotCheck benchmark in Table 2, we set the number of clusters K using the Bayesian information criterion (BIC), following how the SpotCheck benchmark authors chose K for their slice discovery method, PlaneSpot. We will update the draft to include all these details.
>
> We thank you for the feedback, and would be happy to address any other questions that you have.

---

> > ### Comment · Reviewer_6oL7 · 2023-08-14
> > **Response to rebuttal**
> >
> > Thank you for the clarifications. I'm keeping my score as it is.

---

### Official Review · Reviewer_MYaZ · 2023-07-05

**Soundness:** 3 good
**Presentation:** 3 good
**Contribution:** 2 fair
**Rating:** 6
**Confidence:** 3

**Summary:**

This paper proposes a heuristic clustering-based method for identifying errorneous groups of test examples. I'm erring on the side of caution here and go with a weak reject, but I have limited familiarity with the subfield.

AC note: score increased 4 -> 6 after rebuttal.

**Strengths:**

* Error analysis tools are useful for many different ML systems; better algorithms for discovering slices might have have significant practical impact.
* The paper is reasonably clearly written and easy to follow.

**Weaknesses:**

* The main algorithm for estimating the seems to be taken from Schioppa et al.; without it, the methodological contribution of the paper seems quite limited in its nature.
* The paper seems rather simplistic in its approach and not very exciting in its results – cf. Table 2, which is the main evaluation where the technique from the paper is compared to the related work.

**Questions:**

* What is the computational complexity of the method? Is the main computational bottleneck in the implicit Hessian estimation?
* Is there any intuition or an ablation study on the number of elements needed for the Hessian estimation?

**Limitations:**

N\A

---

> ### Author Rebuttal · Authors · 2023-08-10
>
> Thank you for your detailed feedback. We address your concerns below.
>
> **Weaknesses**
> - **Methodological Contribution**: We agree that the actual computation of the influence embeddings being clustered in the main algorithm is a straightforward application of Schioppa et al. [2022]. However, we believe we make 3 key contributions, which we clarify here, and will add to the paper:
> 1. Our work is the first to connect one popular problem in model debugging (slice discovery), with a popular tool in model debugging (influence functions).
> 2. A main contribution is to theoretically derive what the representation to cluster in a slice discovery algorithm should be (influence embeddings), in terms of an influence function-based definition of coherency. This is significant, given that past slice discovery algorithms choose to cluster various representations (CLIP embeddings, last-layer representations, last-layer representations projected by SCVIS) without theoretically explaining why clustering those representations should enable effective slice discovery.  In doing so, we have improved over previously published slice discovery algorithms (e.g. methods that extracted CLIP embeddings or last-layer representations), where the actual computation of those representations did not require any methodological contribution.
> 3. We are the first to use influence functions (of any kind, including but not limited to Schioppa et al) for global explainability (simultaneously explaining predictions for the entire test dataset), unlike past work, which solely focused on local explainability (explaining one prediction at a time, independent of other predictions).
>
> - **Simplicity and Results**: We agree with your take, the algorithm itself is simple; a benefit in practice. However, its development required theoretical insight that past slice discovery approaches missed. In addition, our results (**Table 1 and Table 2**) show that we outperform competing approaches. Lastly, across a series of case studies, we are able to use the scheme to identify underperforming slices in real-world tasks, which is an important challenge in practice. Slice discovery is a critical and challenging task, in practice; the proposed approach is a step towards developing reliable tools for solving that problem.
>
> **Questions**
> - **Computational Complexity**:  Indeed, the main computational bottleneck is the implicit Hessian estimation. Although Schioppa et al. do not explicitly state their complexity, it is O(P), where P is the Arnoldi dimension. This is because each of the P steps in the Arnoldi iteration requires computing a Hessian-vector product, where the Hessian is typically computed on a subset of the training data for tractability. The complexity of computing influence embeddings is then exactly the same as that of the influence function (IF) implementation of Schioppa et al. We also wish to point out the following note in Schioppa et al.’s paper: “Empirically, IFs with Arnoldi iteration achieve speedups of 3-4 orders of magnitude over the LISSA-powered IFs (Koh and Liang 2017) and of 10x over TracIn (Pruthi et al. 2020), a heuristic gradient-only alternative to IFs, with better or similar accuracy.” For the K-means portion, the complexity is O(n_samples$\times$n_iterations$\times$k_clusters). In practice, the clustering step is near instantaneous, so the method is dominated by Hessian estimation.
>
> - **Number of elements needed for hessian estimation**: We estimate the hessian with a random sampling (representative mini-batch) of the training set. In our experiments on Spotcheck and Domino benchmark, we performed ablations to estimate the number of critical samples and found that less than 5 percent of the training samples is needed for such settings. We will update the paper with detailed discussion of these experiments.
>
> We thank you for the feedback, and would be happy to address any other questions that you have. We encourage you to reconsider your score in light of our response.

---

> > ### Comment · Reviewer_MYaZ · 2023-08-14
> >
> > It seems that other reviewers were more excited about paper's contribution than I was. I won't let my lack of enthusiasm hold the paper back.
> > Please add the complexity to the paper! Increasing my score to 6.

---

### Official Review · Reviewer_uzoE · 2023-07-06

**Soundness:** 3 good
**Presentation:** 3 good
**Contribution:** 3 good
**Rating:** 7
**Confidence:** 3

**Summary:**

In this paper, the authors propose InfEmbed on the slice discovery problem. The method is derived from the influence function and surrogate embedding representations are proposed for reducing complexity. Overall, the paper is well written and the derivation of the method is reasonable. Some pros and cons are discussed as follows:

**Strengths:**

The paper proposes a new slice discovery method, which is based on K-Means and the influence embedding the authors proposed. The method is well-designed and I believe is relatively easy for deployment. The overall derivation of the method is clear and theoretical analysis is supported.

The question that the authors are addressing is of importance and the experiment results seem to validate the performance of this method.

The proposed method, if it is effective like the paper states, can provide insights in other domains that require identifying groups of data.


**Weaknesses:**

The K-means is sometimes unstable. As the authors claim the clustering algorithms can be others, they should also validate the results on more cluster algorithms such as the spherical k-means.

The result section is not organized well. It is recommended that the authors discuss datasets, baselines, and results in different subsections and provide more details of data description and baseline methods.


**Questions:**

Do the authors validate the performance of the method with different clustering algorithms?
How is the runtime complexity using influence embedding compared with directly using influence functions?


**Limitations:**

Some results are not well visualized and shown. It is recommened that the authors reorganize the result section to make it more readable.

---

> ### Author Rebuttal · Authors · 2023-08-10
>
> Thank you for your detailed feedback. We address your concerns below.
>
> **Weaknesses**
> - **Trying other clustering algorithms**: We actually did try 3 other clustering algorithms (DBSCAN, spectral clustering, gaussian mixture model) on the Spotcheck benchmark, and found that no matter what clustering algorithm was used on influence embeddings, the performance beat the Domino and PlaneSpot baselines.  These results were unfortunately hidden in Appendix J, and will be moved to the main body.
> - **Organization of results section**: Thank you for the suggestion. We will update the submission to reorganize the results section, and also provide additional overview data and baseline method details.
>
> **Questions**
> - **Trying other clustering algorithms**: Please see the response to weaknesses bullet point 1.
> - **Runtime complexity using influence embedding compared with directly using influence functions**: Directly using influence functions to generate the influence explanation of every test example, and then clustering them would be computationally impractical because for each test example, the length of its influence explanation is the same as the training data size (i.e. if the training data had 1M examples, each test example’s influence explanation would be 1M-dimensional and computing the influence of every training examples on the test example).  In that sense, clustering influence embeddings (instead of influence explanations) is far more efficient, as the dimension of what is being clustered is much smaller.
>
> We would be happy to answer any additional questions that you may have.

---

> > ### Comment · Reviewer_uzoE · 2023-08-20
> >
> > Thanks for the authors' response. I will keep my current ratings.

---

### Official Review · Reviewer_2vT2 · 2023-07-09

**Soundness:** 3 good
**Presentation:** 4 excellent
**Contribution:** 4 excellent
**Rating:** 6
**Confidence:** 3

**Summary:**

This paper proposes a method (InfEmbed) for discovering coherent slices of data such that the model fails on samples within a slice due to similar reasons. InfEmbed uses k-means to cluster a representation proposed in the work called influence embeddings, where samples with similar influence embeddings have similar influence explanations. The proposed slicing method outperforms prior work on standard slice discovery benchmarks.

**Strengths:**

1. The work attempts to solve an important problem of surfacing failure modes of a model automatically.
2. The work has sufficient technical novelty: it proposes influence embeddings, relating them to influence explanations and using them in the context of slice discovery.
3. The work does a thorough evaluation by evaluating the proposed method against multiple slice discovery benchmarks in addition to multiple case studies.


**Weaknesses:**

My main concerns are around hyperparameter selection and missing analysis.
1. The work formalizes coherence (Equation 4), but does not compare to prior works (mainly Domino) in terms of coherence.
2. It is not clear how the method promotes label homogeneity when all gradients of the neural network are considered. The analysis in 3.5 works only when the gradients in the fully connected layer are considered.
3. For a practical use case, the approach still requires the users to determine and specify the branching factor $B$ and the maximum accuracy $A$ of a slice.
4. The work doesn’t provide discussion on the selection of hyperparameter choices: e.g. L231 - the values of P and D, which are used in FactorHessian and the number of clusters ($K$) for the results in Table 1.
5. It is unclear how the authors choose the number [L286] of layers for which gradients are considered. Is it based on the total number of parameters?


**Questions:**

My main concerns are around hyperparameter selection, missing details and analysis.
1. It would be great if the paper can provide error bars in Table 1 for both methods, as Figure 3 of Domino shows high standard deviations across the different settings. Also, it would be useful to report the total number of settings (out of 1235 settings), where InfEmbed outperforms Domino.
2. Can the authors report coherence scores (Equation 4) for different methods in Table 1?
3. Can authors measure the label homogeneity (entropy of predicted distributions) of InfEmbed and compare it with Domino?
4. Can authors provide guidelines on how a practitioner could select the branching factor $B$ and desired accuracy $K$ to get slices with “at most K% accuracy with at least $m$ samples”, which can surface all error modes ranked by accuracy? Currently it seems like the user may have to try a bunch of values to arrive at a reasonable set of slices.
5. What is the value of $K$ used for K-means for the results in Table 1 and Table 2. Does it match the values used by Domino?
6. Section 5: why are the influence embeddings expected to be real valued? In other words, why is  $\lambda$ guaranteed to have positive values?
7. What do the practical values for $C$ in Lemma 1 [L582] look like? Can a theoretical bound be established on the values of $C$?
8. On L188: Why are norm terms treated as a constant?
9. Any reason for marking the reproducibility as N/A?
10. Can the performance be further improved if k-means is replaced with Domino’s error-aware mixture modeling that more directly promotes label homogeneity?



Typos:
* L42: “influence explanation, L
* L215: “whose influence on the slice is most harmful”
* L297: “Apriori one would not have guessed”
* L560: $H_\theta^{P - 1} b$


**Limitations:**

Yes.

---

> ### Author Rebuttal · Authors · 2023-08-10
>
> Thank you for your detailed feedback. We address your concerns below.
>
> **Weaknesses**
> - **Coherence scores and Experiments**: See discussion in the first bullet point of the next section.
> - **How does InfEmbed promote label homogeneity?**: While the analysis in Section 3.5 only considered gradients in the last layer, it does also explain why label homogeneity is encouraged, even when considering gradients in all layers.  This is because for two examples, the dot-product between their gradients in all layers is equal to the dot-product between their gradients in the last-layer, plus the dot-product between their gradients in the remaining layers, and the presence of the former encourages label homogeneity. We will add this clarification to the paper.
> - **How to select B (branching factor) and A (minimum accuracy of a slice) when running InfEmbed-Rule**:
> 1) **Regarding B**: In all experiments, we used branching factor B=3.  The rationale is that B should not be too large, to avoid unnecessarily dividing large slices with sufficient low accuracy into smaller slices.  In practice, B=2 and B=3 did not give qualitatively different results for the case studies where InfEmbed-Rule was used.
> 2) **Regarding A**: The choice of A (seek slices with at most A% accuracy) is application specific, depending on the overall accuracy over the entire test data, and the level of accuracy tolerable for the application. We suggest starting with a low value of A, see if slices are discovered, and increase A if not.  Fortunately, the clustering step is fast, so that this process can be done interactively.
> - **Hyper-parameters (P, D, K)**: Thank you for raising this point. For the case studies, we used a consistent hyper-parameter scheme across the board, and for the baselines, we match the underlying approaches. We will update the draft with all of these hyper-parameters.
> - **Choosing layers**: As is customary in the influence function literature, we choose the last layer. Note that we did not tune this choice for the case study, and it is possible to scale the approach to all parameters for the models we consider.
>
> **Questions**
> - **Error bars, Coherence Scores, and Label Homogeneity**: Thank you for these suggestions. Starting with the domino benchmark, we compute the coherence scores, which corresponds exactly to the K-means objective that we cluster, across all slices for influence embedding clustering as well as for domino. We find a correspondence between our current results (Table 1 in the main paper), and the coherence scores. In every setting where we outperform Domino (1219 out of 1235 trained settings), our coherence scores are better, in some cases, by almost 50 percent. Regarding label homogeneity: In high performing clusters, we find that influence embedding clustering has lower label homogeneity than domino. However, for error clusters we find the reverse. The goal of our scheme is to find clusters where the model is ‘wrong for the same reason’. We conjecture that influence embeddings are most effective for these settings. We will update Table 1 and 2 with error bars, label homogeneity scores, and coherence scores.
> - **Choosing hyper-parameters B, A for InfEmbed-Rule**: Please see response to Weaknesses #3.
> - **Choosing hyperparameter K**: For K (number of slices), we follow past work to ensure comparability of quantitative results: when comparing to Domino in Table 1, we follow Domino and use K=25, and when comparing to various methods on the SpotCheck benchmark in Table 2, we set the number of clusters K using the Bayesian information criterion (BIC), following how the SpotCheck benchmark authors chose K for their slice discovery method, PlaneSpot.
> - **Why are the influence embeddings real-valued?** Thank you for pointing this out. In finding a low-rank approximation of the Hessian, we actually only use the top eigenvectors / eigenvalues for which the eigenvalues are positive. Effectively, this means we first project the Hessian to the closest (in terms of L2 norm) symmetric matrix that is positive definite, before finding its low-rank approximation.  This means that in the algorithm FactorHessian, V, $\lambda** should actually be the top-D eigenvectors / eigenvalues for which the eigenvalues are positive.  We will add this clarification to the paper.
> - **Values of C from the Lemma**: C is the sum of the L2 norms of the influence explanations of the test data, and is thus large.  However, given the strong quantitative results, and that examples within discovered slices tend to have the same opponents, the high value of this theoretical constant does not affect practical results.
> - **Norms**: We apologize for the confusion - what that line is trying to say is that clustering to maximize intra-cluster dot-product similarity is equivalent to clustering to minimize intra-cluster Euclidean distance, because the objective function of the former is equal to the objective function of the latter plus a constant that does not depend on the specific clustering. Therefore, when considering when 2 examples would be placed in the same slice, we just need to consider their dot-product, which is easier to analyze than their Euclidean distance.
> - **Why N/A for reproducibility**: This was a mistake. We will release all our code, datasets, and artifacts to replicate our analyses.
> - **Error aware mixture modeling**: One of the advantages of our method is that by simply applying K-Means to the right representation, influence embeddings, it achieves the same goal as Domino’s error aware mixture model, i.e., the influence embeddings themselves are ‘error-aware’. Having said this, a mixture model can be applied on top of the influence embeddings, and it might be possible to endow the influence embeddings with other qualities. Studying the effects of using influence embeddings with an error aware-mixture model is an interesting future direction.
>
> We would be happy to address additional questions.

---

> > ### Comment · Reviewer_2vT2 · 2023-08-18
> > **Thanks for the rebuttal**
> >
> > I thank the authors for their response. I am updating my rating as I am satisfied with the coherence comparison to prior work and clarifications on hyperparameters. I strongly encourage the authors to include the remaining technical details in the paper/appendix.

---

### Author Rebuttal · Authors · 2023-08-10

**General  Response**\
We thank all the reviewers for their generous feedback, and for noting that the work addresses an important problem (**Reviewers 2vT2, uzoE, yzdi, MYaZ**), is novel (**Reviewers 2vT2, 6oL7**), provides a thorough empirical evaluation (**Reviewers 2vT2, 6oL7, yzdi**), and well-written (**Reviewers MYaZ, 6oL7**).

In addition to the point-by-point response to each reviewer, here, we address some general concerns:

- **Methodological contribution:** A few reviewers asked us to contrast with the closely related work of Schioppa et. al [2022]. We make the following key contributions:
1. We theoretically derive what the representation to cluster in a slice discovery algorithm should be—influence embeddings—in terms of an influence function-based definition of coherency. This is important because past slice discovery algorithms choose to cluster various representations (CLIP embeddings, last-layer representations, last-layer representations projected by SCVIS) without theoretically explaining why clustering those representations should enable effective slice discovery.  In doing so, we have improved over previously published slice discovery algorithms (e.g. methods that extracted CLIP embeddings or last-layer representations).

2. Our work is the first to use influence functions for the slice discovery task. In addition, we are the first to use influence functions (of any kind, including but not limited to Schioppa et. al. 2022) for global explainability (simultaneously explaining predictions for the entire test dataset), unlike past work, which solely focused on local explainability (explaining one prediction at a time, independent of other predictions).

- **Experiments relating to coherence & label homogeneity**: We have now conducted additional experiments to compare the label homogeneity and coherence scores of InfEmbed compared to baselines in several settings. We find that Infembed returns clusters that have higher coherence scores and identifies low-performing clusters that are more label homogenous than baselines.

- **K-means & the importance of the clustering component of Inf-Embed**: In slice discovery, the goal is to partition a set of examples in groups on which a model has high performance versus low performance. So far, this task has been formulated as a search or clustering one. However, the key challenge in this area is to identify a suitable representation that would allow search or clustering algorithms to reliably find low-performing clusters. Consequently, our key contribution is to propose influence embeddings as such a representation for any clustering or search algorithm for slice discovery.

---

### Decision · Program_Chairs · 2023-09-21

**Decision:**

Accept (poster)

**Comment:**

This paper proposes to use k-means clustering influence encodings (InfEmbed) to identify coherent groups of test examples, where the model is wrong (or right) for the same reason. They evaluate on a range of slice discovery benchmarks, including large and diverse datasets such as ImageNet.

Reviewers agree that this is an elegant approach to an important problem, and found the experiments convincing. Initial reviews expressed some concerns about choice of clustering methods, that the methodological contribution was fairly simple, and that the work may be only incremental over Scioppa et al. 2022, but these concerns were largely addressed by the authors' rebuttal, leading three reviewers to raise their scores significantly.